# Have it your way: Individualized Privacy Assignment for DP-SGD

**Franziska Boenisch**[*][†]
boenisch@cispa.de
CISPA Helmholtz Center for Information Security

**Christopher Mühl**[*]
christopher.muehl@fu-berlin.de
Free University Berlin

**Adam Dziedzic**[*][†]
dziedzic@cispa.de
CISPA Helmholtz Center for Information Security

**Roy Rinberg**
roy.rinberg@columbia.edu
Columbia University

**Nicolas Papernot**
nicolas.papernot@utoronto.ca
University of Toronto & Vector Institute

## Abstract

When training a machine learning model with differential privacy, one sets a privacy budget. This uniform budget represents an overall maximal privacy violation that any user is willing to face by contributing their data to the training set. We argue that this approach is limited because different users may have different privacy expectations. Thus, setting a uniform privacy budget across all points may be overly conservative for some users or, conversely, not sufficiently protective for others. In this paper, we capture these preferences through individualized privacy budgets. To demonstrate their practicality, we introduce a variant of Differentially Private Stochastic Gradient Descent (DP-SGD) which supports such individualized budgets. DP-SGD is the canonical approach to training models with differential privacy. We modify its data sampling and gradient noising mechanisms to arrive at our approach, which we call Individualized DP-SGD (IDP-SGD). Because IDP-SGD provides privacy guarantees tailored to the preferences of individual users and their data points, we empirically find it to improve privacy-utility trade-offs.

## 1 Introduction

Machine learning (ML) models are known to leak information about their training data. Such leakage can result in attacks that determine whether a specific data point was used to train a given ML model (membership inference) [36, 43, 8], infer sensitive attributes from the model's training data [13, 45], or even (partially) reconstruct that training data [15, 35]. Therefore, the need to provide privacy guarantees to the individuals who contribute their sensitive data to train ML models is pressing.

Approaches to train ML models with guarantees of differential privacy (DP) [10] have established themselves as the canonical answer to this need. In the context of ML, differential privacy bounds how much can be learned about any of the individual data points from the model's training set. Take the canonical example of differentially private stochastic gradient descent (DP-SGD): it updates the model using noisy averages of clipped per-example gradients. Unlike vanilla SGD, this ensures that any single training point has a limited influence on model updates (due to the clipping to a pre-defined maximum clip norm) and that the training is likely to output a similar model should a single point

---

[*]Equal contribution. [†]Work was done at the University of Toronto and the Vector Institute.

37th Conference on Neural Information Processing Systems (NeurIPS 2023).

be added to or removed from the training set (due to the noisy average). As is the case for any DP algorithm, the privacy guarantees of DP-SGD are derived analytically. They are captured by a privacy budget $\varepsilon$. This budget represents the maximal privacy violation that any user contributing data to the training set is willing to tolerate. Lower budgets correspond to stronger privacy protection since they impose that the outputs of the training algorithm have to be closer for similar training sets.

In this paper, we argue that there is a key limitation when training ML models with DP guarantees: the privacy budget $\varepsilon$ is set uniformly across all training points. This implicitly assumes that all users who contribute their data to the training set have the same expectations of privacy. However, that is not true; individuals have diverse values and privacy preferences [17, 5]. Yet, because DP-SGD assumes a uniform privacy budget, this budget must match the individual whose privacy expectations are the strongest. That is, $\varepsilon$ has to correspond to the lowest privacy violation tolerance expressed amongst individuals who contributed their data. This limits the ability of DP-SGD to learn because the privacy budget $\varepsilon$ together with the sensitivity of each step of the algorithm (*i.e.*, the clip norm) define the scale of noise that needs to be added: the stronger the required privacy protection, the higher the noise scale. Thus, implementing privacy protection according to the lowest privacy violation tolerance in the training dataset comes at a significant cost in model accuracy.

We are the first to propose two variants of the DP-SGD algorithm, that we refer to as IDP-SGD, to enforce different privacy budgets for each training point.[2] First, our modified sampling mechanism (Sample) samples data points proportionally to their privacy budgets: points with higher privacy budgets are sampled more frequently during training. It is intuitive that the more a training point is analyzed during training, the more private information can leak about this point. Second, with our individualized scaling mechanism (Scale), we rescale the noise added to gradients based on the budget specified for each training point. Naively, one would implement Scale by changing the scale of added noise per individual data point. However, for efficiency, current implementations of DP-SGD typically noise the *sum* of the per-example clipped gradients over an entire mini-batch, hence, adding the same amount of noise to all gradients. Yet, within the context of this implementation, we note that we can *effectively* adapt the scale of noise being added on an individual basis by adjusting the sensitivity (*i.e.* the clip norm) of each example by a multiplier. Because clipping is performed on a per-example basis, this approach enables the efficient implementation of our proposed Scale method.

To summarize our contributions:

- We introduce two novel individualized variants of the DP-SGD algorithm, where each data point can be assigned its desired privacy budget. This allows individuals who contribute data to the training set to specify different privacy expectations.
- We provide a theoretical analysis of our two mechanisms to derive how the privacy parameters (*e.g.*, noise multipliers and sample rates) should be set to meet individual data points' privacy guarantees and provide a thorough privacy analysis for Sample and Scale.
- We carry out an extensive empirical evaluation on various vision and language datasets and multiple model architectures. Our results highlight the utility gain of our methods for different distributions of privacy expectations among users.

**Ethical considerations.**    We need to ensure that our individualized privacy assignment does not harm individuals by disclosing too much of their sensitive information. We, therefore, suggest implementing our methods in a controlled environment where privacy risks are openly communicated [41, 9, 12], and where a regulatory entity is in charge of ensuring that even the individuals with lowest expectations regarding their privacy obtain a sufficient degree of protection. We further discuss the ethical deployment of our Individualized DP-SGD (IDP-SGD) in Appendix A.

## 2   Background and Related Work

### 2.1   Differential Privacy and DP-SGD

Algorithm $M$ satisfies $(\varepsilon, \delta)$-**DP**, if for any two datasets $D, D' \subseteq \mathcal{D}$ that differ in any one record $x$, such that $D = D' \cup x$, and any set of outputs $R$

$$\mathbb{P}\left[M(D) \in R\right] \leq e^{\varepsilon} \cdot \mathbb{P}\left[M(D') \in R\right] + \delta. \tag{1}$$

---

[2]In the rest of this paper, we assume without loss of generality that each training point is contributed by a different individual.

Since the possible difference in outputs is bounded for any pair of datasets, DP bounds privacy leakage for any individual. The value $\varepsilon \in \mathbb{R}_+$ specifies the privacy level with lower values corresponding to stronger privacy guarantees. The $\delta \in [0, 1]$ offers a relaxation, *i.e.*, a small probability of violating the guarantees. See Appendix B.1 for a more thorough introduction of $(\varepsilon, \delta)$-DP.

Another relaxation of DP is based on the Rényi divergence [27], namely $(\boldsymbol{\alpha}, \boldsymbol{\varepsilon})$**-RDP**, which for the parameters as described above and an order $\alpha \in (1, \infty)$ is defined as follows: $\mathbb{D}_\alpha \left( M(D) \parallel M(D') \right) \leq \varepsilon$. Due to its smoother composition properties, it is employed for privacy accounting in ML with DP-SGD. Yet, it is possible to convert between the two notions: If $M$ is an $(\alpha, \varepsilon)$-RDP mechanism, it also satisfies $(\varepsilon + \log \frac{\alpha-1}{\alpha} - \frac{\log \delta + \log \alpha}{\alpha-1}, \delta)$-DP for any $0 < \delta < 1$ [3].

**DP-SGD** extends standard SGD with two additional steps, namely clipping and noise addition. Clipping each data point's gradients to a pre-defined *clip norm* $c$ bounds their *sensitivity* to ensure that no data point causes too large model updates. Adding noise from a Gaussian distribution, $\mathcal{N}(0, (\sigma c)^2 \mathbf{I})$, with zero mean and standard deviation according to the sensitivity $c$ and a pre-defined *noise multiplier* $\sigma$ introduces privacy. We formally depict DP-SGD as Algorithm 1. To yield tighter privacy bounds, DP-SGD implements privacy amplification by subsampling [4] (line 3 in Algorithm 1) where training data points are sampled into mini-batches with a Poisson sampling,[3] rather than assigning each data point from the training dataset to a mini-batch before an epoch starts. DP-SGD hence implements a Sampled Gaussian Mechanism

---

**Algorithm 1:** Differentially Private SGD [1]

**Require:** Training dataset $\mathcal{D}$ with data points $\{x_1, \ldots, x_N\}$, loss function $l$, learning rate $\eta$, noise multiplier $\sigma$, sampling rate $q$, clip norm $c$, number of training iterations $I$.
1: **Initialize** $\theta_0$ randomly
2: **for** $t \in [I]$ **do**
3:     *Poisson Sampling*: Sample mini-batch $L_t$ with per-point probability $q$ from $\mathcal{D}$.
4:     For each $i \in L_t$, compute $g_t(x_i) \leftarrow \nabla_{\theta_t} l(\theta_t, x_i)$
5:     **1. Gradient Clipping**
6:     $\bar{g}_t(x_i) \leftarrow g_t(x_i) / \max \left( 1, \frac{\|g_t(x_i)\|_2}{c} \right)$
7:     **2. Noise Addition**
8:     $\tilde{g}_t \leftarrow \frac{1}{|L_t|} \left( \sum_i \bar{g}_t(x_i) + \mathcal{N}(0, (\sigma c)^2 \mathbf{I}) \right)$
9:     $\theta_{t+1} \leftarrow \theta_t - \eta \tilde{g}_t$
10: **end for**
11: **Output** $\theta_I$, privacy cost $(\varepsilon, \delta)$ computed using a privacy accounting method.

---

(SGM) [28] where each training data point is sampled independently at random without replacement with probability $q \in (0, 1]$ over $I$ training iterations. As shown by Mironov *et al.* [28], an SGM with sensitivity of $c = 1$, satisfies $(\alpha, \varepsilon)$-RDP where

$$\varepsilon \leq I \cdot 2q^2 \frac{\alpha}{\sigma^2}. \tag{2}$$

Equation (2) highlights that the privacy guarantee $\varepsilon$ depends on the noise multiplier $\sigma$, the sample rate $q$, the number of training iterations $I$, and the RDP order $\alpha$.

## 2.2 Individualized Privacy

**Individualized Privacy** is a topic of high importance given that society consists at least of three different groups of individuals, demanding either strong, average, or weak privacy protection for their data, respectively [17, 5]. Furthermore, some individuals inherently require higher privacy protection, given their personal circumstances and the degree of uniqueness of their data. However, without individualization, when applying DP to datasets that hold data from individuals with different privacy requirements, the privacy level $\varepsilon$ has to be chosen according to the lowest $\varepsilon$ encountered among all individuals, which favors poor privacy-utility trade-offs. Prior work on individualized privacy guarantees focused mainly on standard data analyses without considering ML [2, 19, 24, 31]. However, some of the underlying ideas inspired the design of our IDP-SGD variants.

Our **Sample** method relies on a similar idea as the *sample mechanism* proposed by Jorgensen et al. [19] where data points with stronger privacy requirements are sampled with a lower probability than data points that agree to contribute more of their information. The most significant difference between the *sample mechanism* and our approach is that the former performs sampling as a pre-processing step independent of the subsequent algorithm. In contrast, we perform subsampling *within* every iteration of our IDP-SGD training and show how to leverage the sampling probability of DP-SGD to obtain individual privacy guarantees. Based on this difference, we could not build on how Jorgensen

---

[3]In DP-SGD, the concept of an epoch does not exist and training duration is measured in *iterations*. Due to the Poisson sampling, training mini-batches can contain varying numbers of data points.

et al. [19] compute the sampling probabilities and had to propose an original mechanism to derive sampling probabilities for individualized privacy in IDP-SGD.

In a similar vein to our **Scale** method, Alaggan et al. [2] scale up data points individually before the noise addition to increase their sensitivity in their *stretching mechanism*. As a consequence, the signal-to-noise ratios of data points that are scaled with large factors will be higher than the ones of data points that are scaled with small factors. In contrast to Alaggan et al. [2], we do not scale the data points themselves, but the magnitude of noise added to their gradients, relative to their individual sensitivity. Our new interesting observation is that data points' individual clip norms can be used to indirectly scale the noise added to a mini-batch of data in DP-SGD per data point.

Closest work to ours [6] proposes two mechanisms for an individualized privacy assignment within the framework of Private Aggregation of Teacher Ensembles (PATE) [33]. Their *upsampling* duplicates training data points and assigns the duplicates to different teacher models according to the data points' privacy requirements, while their *weighting* changes the aggregation of the teachers' predicted labels to reduce or increase the contribution of teachers according to their respective training data points' privacy requirements. PATE's approach for implementing ML with DP differs significantly from DP-SGD: PATE relies on privately training multiple non-DP models on the sensitive data and introducing DP guarantees during a knowledge transfer to a separately released model. The approaches for individualized privacy in PATE, therefore, are non-applicable to DP-SGD. See Appendix B.3 for details on the PATE algorithm and its individualized extensions.

**Relationship to Individualized Privacy Accounting.** An orthogonal line of research that comes closest to providing individualized guarantees for DP-SGD focuses on individualized *privacy accounting*; the idea is to learn more from points that consume less of the privacy budget over training [11, 44]. Jordon et al. [18] propose a personalized moments' accountant to compute the privacy loss on a per-sample basis. Yu et al. [44] perform individualized privacy accounting within DP-SGD based on the gradient norms of the individual data points. Feldman and Zrnic [11] introduce *RDP filters* to account for privacy consumption per data point and train as long as there remain data points that do not exceed the global privacy level. Yet, it has been shown that the data points that consume little privacy budget and therefore remain in the training are the ones that already incur a small training loss [44]. Training more on them will not significantly boost the model accuracy. While privacy assignment and accounting are concerned with improving privacy-utility trade-offs within DP, they operate under different setups. Privacy *accounting* (in the above three methods) assumes a single fixed privacy budget assigned over the whole dataset. In contrast, our privacy *assignment* is concerned with enabling different individuals to specify their respective privacy preferences. Since individual accounting and assignment are two independent methods, we experimentally show their synergy by applying individual assignment to individual accounting [11], which yields higher model utility than individual accounting alone.

## 3 Our Individualized Privacy Assignment

### 3.1 Formalizing Individualizing Privacy

**Setup and Notation.** Given a training dataset $D$ with points $\{x_1, \ldots, x_N\}$ which each have their own privacy preference (or *budget*), we consider data points with the same privacy budget $\varepsilon_p$ together as a *privacy group* $\mathcal{G}_p$. This notation is aligned with [17, 5] that identified different *groups* of individual privacy preferences within society. We denote with $\varepsilon_1$ the smallest privacy budget encountered in the dataset. Standard DP-SGD needs to set $\varepsilon = \varepsilon_1$ to comply with the strongest privacy requirement encountered within the data. For all $\varepsilon_p, p \in [2, P]$, it follows that $\varepsilon_p > \varepsilon$ and we arrange the groups such that $\varepsilon_p > \varepsilon_{p-1}$. Note that following [2], we argue that the privacy preferences themselves should be kept private to prevent leakage of sensitive information which might be correlated with them. If there is a necessity to release individual privacy preferences, this should be done under the addition of noise to obtain DP guarantees for the privacy preferences. This can for instance be done through a smooth sensitivity analysis, as done in prior work, *e.g.* [34]. In general, we find that it is not necessary to release the privacy budgets: To implement IDP-SGD, only the party who trains the ML model on the individuals' sensitive data needs to know their privacy budgets to set the respective privacy parameters accordingly during training. Given that this party holds access to the data itself, this does not incur any additional privacy disclosure for the individuals. Other parties then only interact with the final private model (but not with the individuals' sensitive data or their

associated privacy budgets) and it has been shown impractical to disclose privacy guarantees through black-box access to ML models [14]. We discuss the confidentiality of the privacy budgets further in Appendix A.2.

**Individualized Privacy.** Following [19], our methods aim at providing data points $x_i \in \mathcal{G}_p$ with individual DP (IDP) guarantees according to their privacy budget $\varepsilon_p$ as follows: The learning algorithm $M$ satisfies $(\varepsilon_p, \delta)$-IDP if for all datasets $D \overset{x_i}{\sim} D'$ which differ only in $x_i$ and for all outputs $R \subseteq \mathcal{R}$

$$\mathbb{P}\left[M(D) \in R\right] \leq e^{\varepsilon_p} \cdot \mathbb{P}\left[M(D') \in R\right] + \delta. \tag{3}$$

Without loss of generality, we assume that all individual data points have privacy preferences with the same $\delta$. Note that other notions of DP (*e.g.*, RDP) can analogously be generalized to enable per-data point privacy guarantees. The key difference between Equation (3) and the standard definition of DP (Equation (1)) does not lie in the replacement of $\varepsilon$ with $\varepsilon_p$. Instead, the key difference is about the definition of neighboring datasets. For standard DP, neighboring datasets are defined as $D \sim D'$ where $D$ and $D'$ differ in any random data point (given that it is standard DP, every data point has privacy $\varepsilon$). In contrast, in our work, following [19] the neighboring datasets are defined as $D \overset{x_i}{\sim} D'$ where $D$ and $D'$ differ in any arbitrary data point $x_i$ which has a privacy budget of $(\varepsilon_p, \delta)$. This yields to the individualized notion of DP.

We formalize the relationship between the individualized notion of DP (which we denote by $(\{\varepsilon_1, \varepsilon_2, \ldots, \varepsilon_P\}, \delta)$-IDP) and standard $(\varepsilon, \delta)$-DP in the following two lemmas over the learning algorithm $M$. The proofs are included in Appendix G.

**Lemma 3.1.** *An algorithm $M$ that satisfies $(\varepsilon_1, \delta)$-DP also satisfies $(\{\varepsilon_1, \varepsilon_2, \ldots, \varepsilon_P\}, \delta)$-IDP.*

**Lemma 3.2.** *An algorithm $M$ that satisfies $(\{\varepsilon_1, \varepsilon_2, \ldots, \varepsilon_P\}, \delta)$-IDP also satisfies $(\varepsilon_P, \delta)$-DP.*

## 3.2 From Individualized Privacy Preferences to Privacy Parameters

From the individual privacy preferences and the total given privacy distribution over the data, *i.e.*, the number of different privacy budgets $P$, their values $\varepsilon_p$, and the sizes of the respective privacy groups $|\mathcal{G}_p|$, we derive individual privacy parameters for IDP-SGD such that all data points within one privacy group (same privacy budget) obtain the same privacy parameters, and such that all privacy groups are expected to exhaust their privacy budget at the same time, after $I$ training iterations.

The individualized parameters for our Sample method are the individual sample rates $\{q_1, \ldots, q_P\}$, and a respective noise multiplier $\sigma_{\text{sample}}$—common to all privacy groups—which we derive from the privacy budget dis-

Table 1: **(Individualized) Privacy Bounds.**

| DP-SGD | SAMPLE | SCALE |
|---|---|---|
| $\varepsilon \leq I \cdot 2q^2 \frac{\alpha}{\sigma^2}$ | $\varepsilon_p \leq I \cdot 2q_p^2 \frac{\alpha}{\sigma_{sample}^2}$ | $\varepsilon_p \leq I \cdot 2q^2 \frac{\alpha}{\sigma_p^2}$ |

tribution as described in Section 3.3. In Scale, the individualized parameters are noise multipliers $\{\sigma_1, \ldots, \sigma_P\}$ with their respective clip norms $\{c_1, \ldots, c_P\}$, as we explain in Section 3.4. Our individual bounds per privacy group are depicted in Table 1. We omit $\alpha$ from consideration since its optimal value is selected as an optimization parameter when converting from RDP to $(\varepsilon, \delta)$-DP.

## 3.3 Our Sample Method

Our Sample method relies on sampling data points with different sample rates $\{q_1, \ldots, q_P\}$ depending on their individual privacy budgets. In this case, the noise multiplier $\sigma_{\text{sample}}$ is fixed. Data points with higher privacy budgets (weaker privacy requirements) are assigned higher sampling rates than those with lower privacy budgets. This modifies the Poisson sampling for DP-SGD (line 3 in Algorithm 1) to sample data points with higher privacy budgets within more training iterations.

**Deriving Parameters.** We aim at deriving the privacy parameters that yield the specified individual privacy preferences. The learning hyperparameters, such as mini-batch size $B$ or learning rate $\eta$ can be found through hyperparameter tuning.[4] For Sample, given a tuned mini-batch size $B$, we have to find $\{q_1, \ldots, q_P\}$, such that their weighted average equals $q$: $\frac{1}{N} \sum_{p=1}^{P} |\mathcal{G}_p| q_p \overset{!}{=} q = \frac{B}{N}$.[5]

---

[4]We discuss in Appendix C.1 other alternatives to implement Sample if we allow changing the *training* hyperparameters.

[5]We use the notation $\overset{!}{=}$ to indicate that the two terms *should be* equal.

**Algorithm 2: Finding Sample Parameters.**
The subroutine *getSampleRate* is equivalent to Opacus' function get_noise_multiplier [32], (see *getNoise* inAlgorithm 5). We sketch *getSampleRate* in Algorithm 4 in Appendix C.3.

**Require:** Per-group target privacy budgets $\{\varepsilon_1, \ldots, \varepsilon_P\}$, target $\delta$, iterations $I$, number of total data points $N$ and per-privacy group data points $\{|\mathcal{G}_1|, \ldots, |\mathcal{G}_P|\}$.
1: **init** $\sigma_{\text{sample}}$: $\sigma_{\text{sample}} \leftarrow \text{getNoise}(\varepsilon_1, \delta, q, I)$
2: **init** $\{q_1, \ldots, q_P\}$ where for $p \in [P]$:
3: $\quad q_p \leftarrow \text{getSampleRate}(\varepsilon_p, \delta, \sigma_{\text{sample}}, I)$
4: **while** $q \not\approx \frac{1}{N} \sum_{p=1}^{P} |\mathcal{G}_p| q_p$:
5: $\quad \sigma_{\text{sample}} \leftarrow s_i \sigma_{\text{sample}}$ {scaling factor: $s_i < 1$}
6: $\quad$ **for** $p \in [P]$:
7: $\quad\quad q_p \leftarrow \text{getSampleRate}(\varepsilon_p, \delta, \sigma_{\text{sample}}, I)$
8: **Output** $\sigma_{\text{sample}}, \{q_1, \ldots, q_P\}$

**Algorithm 3: Finding Scale Parameters.**
The subroutine *getNoise* corresponds to Opacus' function get_noise_multiplier [32]. We sketch its implementation in Algorithm 5 in Appendix C.3.

**Require:** Per-group target privacy budgets $\{\varepsilon_1, \ldots, \varepsilon_P\}$, target $\delta$, iterations $I$, number of total data points $N$ and per-privacy group data points $\{|\mathcal{G}_1|, \ldots, |\mathcal{G}_P|\}$, default clip norm $c$, sample rate $q$.
1: **init** $\{\sigma_1, \ldots, \sigma_P\}$ where for $p \in [P]$:
2: $\quad \sigma_p \leftarrow \text{getNoise}(\varepsilon_p, \delta, q, I)$
3: **init** $\sigma_{\text{scale}}$: $\sigma_{\text{scale}} \leftarrow (\frac{1}{N} \sum_{p=1}^{P} \frac{|\mathcal{G}_p|}{\sigma_p})^{-1}$ {see derivation of $\sigma_{\text{scale}}$ in Appendix C.2}
4: **for** $p \in [P]$:
5: $\quad c_p \leftarrow \frac{\sigma_{\text{scale}} c}{\sigma_p}$
6: **Output** $\sigma_{\text{scale}}, \{c_1, \ldots, c_P\}$

This asserts that the Poisson sampling (line 3 in Algorithm 1) will yield mini-batches of size $B$ in expectation. We also need to ensure that the privacy budgets of all groups will exhaust after $I$ training iterations, which we do by deriving the adequate $\sigma_{\text{sample}}$, shared across all privacy groups. Equation (2) highlights that the final privacy budget $\varepsilon$ depends on both sample rate $q$ and noise multiplier $\sigma$. Our Sample method enables higher individualized privacy budgets for the different groups than the standard DP-SGD but still requires setting the average sampling rate equal to $q$ as in DP-SGD to obtain an expected mini-batch size $B$. Hence, the value of $\sigma_{\text{sample}}$ has to be decreased in comparison to the initial default $\sigma$. As an effect of this noise reduction, Sample improves utility of the final model.

Our concrete derivation of values for $\sigma_{\text{sample}}$ and $\{q_1, \ldots, q_P\}$ is presented in Algorithm 2. We start from initializing $\sigma_{\text{sample}}$ with $\sigma$ from standard DP-SGD, the noise multiplier required for the privacy group $\mathcal{G}_1$ (strongest privacy requirement of all groups) with the smallest privacy budget $\varepsilon_1$. This corresponds to instantiating $\sigma_{\text{sample}}$ with the upper bound noise over all privacy groups. Then, we use a *getSampleRate* function that derives the sampling rate for the given privacy parameters based on approximating Equation (2). Finally, we iteratively decrease $\sigma_{\text{sample}}$ by a scaling factor $s_i$ slightly smaller than one and recompute $\{q_1, \ldots, q_P\}$ until their weighted average is (approximately) equal to $q$. In Appendix G, we present the formal proof that Sample satisfies $(\{\varepsilon_1, \varepsilon_2, \ldots, \varepsilon_P\}, \delta)$-DP. Our proof relies on Mironov et al. [28] and considers training of each privacy group as a separate SGM—with an individual sampling probability—that all simultaneously update the same model.

### 3.4 Scale

Our Scale method aims at scaling the noise added to each gradient according to the respective data point's privacy preference. Yet, for efficiency, current implementations of DP-SGD typically noise the *sum* of the per-example clipped gradients over an entire mini-batch, hence, adding the same amount of noise to all gradients. Therefore, we instead set individualized clip norms $\{c_1, \ldots, c_P\}$ that *effectively* adapt the scale of noise being added on an individual basis by adjusting the sensitivity (*i.e.* the clip norm) of each example by a multiplier. This changes lines 6 and 8 in the DP-SGD Algorithm (1). Data points with higher privacy budgets (weaker privacy requirements) obtain lower noise and higher clip norms. Since Equation (2) highlights that the clip norm $c$ has no direct impact on the obtained $\varepsilon$, individualized privacy in Scale results from the individual noise multipliers $\{\sigma_1, \ldots, \sigma_P\}$. Utility gains come from the overall increase in the signal-to-noise ratio during training.

**Deriving Parameters.** While the ultimate goal of our Scale method is to adapt individual noise multipliers per privacy group, we cannot implement this directly without degrading training performance. The reason is that whereas in DP-SGD sampling and gradient clipping are performed on a per-data point basis, noise is added per mini-batch, see lines 3, 6 and 8 in Algorithm 1, respectively. However, if we restrict mini-batches to contain only data points from the same privacy group (which

share the same noise multiplier), we lose the gains in privacy-utility trade-offs which result from the subsampling (see Appendix F.1 for more details). Hence, while we rely on mini-batches containing data points with different privacy requirements (*i.e.*, different noise multipliers) we can only specify one fixed noise multiplier $\sigma_{\text{scale}}$.

To overcome this limitation, we do not set noise multipliers $\{\sigma_1, \ldots, \sigma_P\}$ directly, but indirectly obtain them through individualized clip norms $\{c_1, \ldots, c_P\}$ as follows: In standard DP-SGD, Algorithm 1, a gradient clipped to $c$ (line 6) obtains noise with standard deviation $\sigma c$ (line 8). For Scale, we clip gradients to $c_p = s_p c$ with a per-privacy group scaling factor $s_p$ and they obtain noise $\sigma_p c_p$. But in practice, we add noise according to $\sigma_{\text{scale}} c$ to all mini-batches. Hence, the effective scale $\sigma_p$ of added noise is $\sigma_{\text{scale}} c = \sigma_{\text{scale}} \frac{c_p}{s_p} \stackrel{!}{=} \sigma_p c_p \Rightarrow \sigma_p = \frac{1}{s_p} \sigma_{\text{scale}}$. For data points with higher privacy budgets $s_p > 1$, so their gradients are clipped to larger norms $c_p = s_p c$ and a smaller noise multiplier $\sigma_p = \frac{1}{s_p} \sigma_{\text{scale}}$ is assigned to them. The opposite is true for data points with lower privacy budgets.

We find the values of $\sigma_p$ required to obtain each privacy groups' desired $\varepsilon_p$ using the *getNoise* function (see Algorithm 3 line 1), which takes as inputs the privacy parameters $\varepsilon_p, \delta, q, I$. To optimize utility, we want to set the individual clip norms such that their average over the dataset corresponds to the best clip norm $c$ obtained through initial hyperparameter tuning: $\frac{1}{N} \sum_{p=1}^{P} |\mathcal{G}_p| c_p \stackrel{!}{=} c$. Given the interdependence of $c_p, \sigma_p$, and $\sigma_{\text{scale}}$ ($c_p = c \frac{\sigma_{\text{scale}}}{\sigma_p}$), this can be achieved by setting $\sigma_{\text{scale}}$ as the inverse of the weighted average over all $1/\sigma_p$ as: $\sigma_{\text{scale}} = (\frac{1}{N} \sum_{p=1}^{P} \frac{|\mathcal{G}_p|}{\sigma_p})^{-1}$ (please see Appendix C.2 for the derivation of $\sigma_{\text{scale}}$). Given $\sigma_{\text{scale}}$, the required $\{\sigma_1, \ldots, \sigma_P\}$, and $c$ found through hyperparamter tuning, the individual clip norms are calculated as $c_p = \frac{\sigma_{\text{scale}} c}{\sigma_p}$. We detail the derivation of the parameters in Algorithm 3. In Appendix G, we formally show that Scale satisfies $(\{\varepsilon_1, \varepsilon_2, \ldots, \varepsilon_P\}, \delta)$-DP). Similar to Sample, the proof is based on considering the training for all privacy groups as simultaneously executed SGMs with differing sensitivities.

## 4 Empirical Evaluation

For our empirical evaluation, we implement our methods in Python 3.9 and extend standard Opacus with our individualized privacy parameters and a per-privacy group accounting. We perform evaluation on the MNIST [23], SVHN [29], and CIFAR10 [22] dataset, using the convolutional architectures from Tramer and Boneh [38] for most experiments, and additionally evaluate on various language datasets and diverse (larger) model architectures in Section 4.2. To evaluate the utility of our methods, we use the datasets' standard train-test splits and report test accuracies. The training and standard DP-SGD and IDP-SGD hyperparameters are specified in Table 5 and Table 6 in Appendix D, respectively where the noise multiplier $\sigma$ is derived with the function `get_noise_multiplier` provided in Opacus [32]. It takes in as arguments the specified parameters $\delta, q, I$, and target $\varepsilon$. For experimentation on individualized privacy, we assigned privacy budgets randomly to the training data points if not indicated otherwise.

### 4.1 Utility Improvement and General Applicability of Individualization

Assigning heterogeneous individual privacy budgets over the training dataset yields significant improvements in terms of the resulting model's utility, as showcased in Table 2. For example, by following the privacy budget distribution of Alaggan et al. [2] with privacy budgets $\varepsilon_p = \{1, 2, 3\}$ for strong, medium, and weak privacy requirements, our Sample method yields accuracy improvements of 1.06%, 2.63%, and 5.09% on MNIST, SVHN, and CIFAR10, respectively. On the CIFAR10 dataset, our Scale even outperforms improvement with an accuracy increase of 5.26% in comparison to the non-individualized baseline, which would have to assign $\varepsilon = 1$ to the whole training set in order to respect each individual's privacy preferences.

The benefits of our individualized privacy assignment also become clear in Figure 1, which depicts the test accuracy of our Sample and Scale vs. standard DP-SGD over the course of training. Both our methods continuously outperform the non-individualized DP-SGD baseline with $\varepsilon = 1$. Additionally, the test accuracy with privacy budget distribution according to Alaggan et al. [2] $(34\%, 43\%, 23\%)$ is higher than the one of Niu et al. [30] $(54\%, 37\%, 9\%)$. This can be explained by the fact that in this latter distribution, more individuals exhibit a higher pri-

Table 2: **Model Test Accuracy after training with Standard DP-SGD vs our Individualized DP-SGD** using Sample or Scale. The percentages of the three privacy groups are chosen according to [2] (first setup) and [30] (second setup). Additional hyperparameters are found in Table 5 and Table 6. We report the standard deviation over 10 trials.

| DATASET | PRIVACY GROUPS | PRIVACY BUDGETS | DP-SGD | SAMPLE | SCALE |
|---|---|---|---|---|---|
| MNIST | *34%-43%-23%* | *1.0-2.0-3.0* | 96.75±0.15 | **97.81±0.09** | 97.78±0.08 |
| | *54%-37%-9%* | *1.0-2.0-3.0* | 96.75±0.15 | **97.6±0.11** | 97.54±0.09 |
| SVHN | *34%-43%-23%* | *1.0-2.0-3.0* | 83.26±0.31 | **85.89±0.14** | 85.57±0.24 |
| | *54%-37%-9%* | *1.0-2.0-3.0* | 83.26±0.31 | **85.14±0.30** | 85.08±0.12 |
| CIFAR10 | *34%-43%-23%* | *1.0-2.0-3.0* | 52.77±0.65 | 57.86±0.56 | **58.03±0.36** |
| | *54%-37%-9%* | *1.0-2.0-3.0* | 52.77±0.65 | **56.83±0.39** | 56.65±0.49 |

vacy preference. With more data points choosing lower privacy protection as in Alaggan et al. [2], our individualization boosts the performance of the final trained model more significantly.

For comparison, we also report results for the standard DP-SGD with $\varepsilon = 3$. This corresponds to the upper bound on utility that could have been achieved by assigning the highest privacy budget to all data points—which violates the privacy requirements of individuals with strong and medium privacy preferences. We observe that our individualized methods' performance is close to this upper bound without violating data points' privacy requirements. In Appendix E, we also discuss other possible baselines (apart from *standard* DP-SGD) and empirically evaluate our indi-

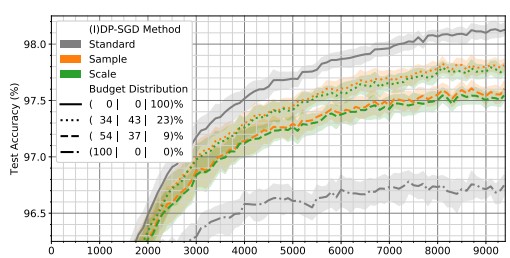

Figure 1: **IDP-SGD vs DP-SGD.**

vidualized methods against them. We report our hyperparameters and the respective individualization parameters (noise multipliers, clip norms, and sampling rates) obtained through our methods in Appendix D. Figure 4 in Appendix D.1 highlights additionally that this privacy parameter derivation is well calibrated: all different groups exhaust their privacy budget simultaneously when the targeted number of iterations is reached. In Appendix D.2, we also experimentally show that our methods are flexible and can extend to substantially more privacy groups which all have different budgets.[6] Finally, in Appendix D.6, we demonstrate how to combine our Sample and Scale method, and experimentally evaluate the joint method.

## 4.2 Broad Applicability of IDP-SGD

We also evaluate IDP-SGD with other (larger) architectures, for fine-tuning, and with different modalities and tasks. Therefore, we 1) fine-tune a BERT transformer [21] (bert-base-uncased) on the SNLI dataset [7] for natural language inference, 2) train a ResNet18 [16] from scratch on the CIFAR10 dataset for image classification, 3) train a linear embedding model (one embedding and two linear layers) on IMDB data [25] for text classification, and 4) train a character-level RNN (DPLSTM[7], three layers, hidden size: 128) from scratch to classify surnames to languages. In Table 3, we evaluate our Sample and Scale in these setups with a privacy distribution corresponding to the first setup of Table 2, with $\varepsilon = 5, 7.5, 10$ for the first three rows, and $\varepsilon = 1, 2, 3$ for the last one, and a DP-SGD baseline with $\varepsilon = 5$ or $\varepsilon = 1$, respectively. We tuned hyperparameters with standard DP-SGD and applied the best resulting hyperparameters to Sample and Scale. All results are averaged over 9 trials and we report average accuracy and standard deviation. Over all tasks, architectures, and modalities, IDP-SGD outperforms standard DP-SGD.

## 4.3 Privacy Assessment via Membership Inference

To evaluate the impact of our IDP-SGD on the privacy of individual data points, we perform membership inference based on the LiRA attack [8]. For the reader's convenience, we include a

---

[6]In principle, with our methods, each data point could have its own privacy budget.
[7]https://opacus.ai/tutorials/building_lstm_name_classifier

Table 3: **Evaluating IDP-SGD with other architectures, tasks, modalities, and to fine-tuning.**

| Architecture | Dataset | Setup | Modality, Task | DP-SGD | Sample | Scale |
|---|---|---|---|---|---|---|
| BERT | SNLI | Fine-Tune | Natural language inference | 75.91± 0.23 | 76.11±0.21 | **76.5±0.17** |
| ResNet18 | CIFAR10 | Train | Image classification | 47.52±0.84 | 48.53±0.69 | **48.77±0.73** |
| Embedding Model | IMDB | Train | Text classication | 72.69±0.27 | 73.27±0.3 | **73.34±0.11** |
| Character-level RNN | Surnames | Train | Text (name) classification | 60.86±0.78 | 65.56±0.96 | **66.0±1.19** |

description of the attack in Appendix B.5. We experiment with CIFAR10 and train 512 shadow models and the target model using Sample on different subsets of 25,000 training data points each. Results for Scale can be found in Appendix D.3. The privacy budgets are set to $\varepsilon = 10$ and $\varepsilon = 20$ and evenly assigned to the shadow models' training data, resulting in 12,500 training data points per privacy budget. Our target model achieves a train accuracy of 68.46% on its 25,000 member data points, and a test accuracy of 64.89% on its 25,000 non-member data points. The lower accuracy in comparison to Carlini et al. [8], who achieved 100% and 92% train and test accuracy respectively, results from us introducing DP to the training of the shadow models. Learning with DP is known to reduce model performance, particularly so on small datasets.

We depict the membership inference risk of the target model's training data per privacy budget (privacy group) in Figure 2. The dotted blue line represents the ROC curve for LiRA over all CIFAR10 data points and shows that overall, as expected by its definition, DP protects the training data well against membership inference attacks. However, when inspecting the ROC curve separately for the two privacy groups ($\varepsilon = 10$ and $\varepsilon = 20$), we observe a significant difference between them. The privacy group with stronger privacy guarantees ($\varepsilon = 10$) is protected better (AUC= 0.537) than the group with the higher privacy budget ($\varepsilon = 20$, AUC= 0.581). To evaluate whether the difference in the LiRA-likelihood scores between the two privacy groups is statistically significant, we perform a Student t-test on the likelihood score distributions over the data points with privacy budget $\varepsilon = 10$ vs. the data points with privacy budget $\varepsilon = 20$. We report $\Delta = 2.54$ with $p = 0.01 < 0.05$, and hence a statistically significant difference between the two groups exists.

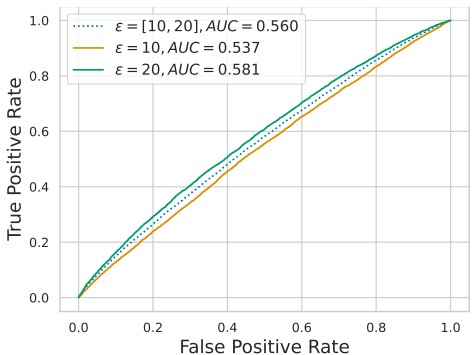

Figure 2: **Per-Privacy Group LiRA [8] Membership Inference Success.**

Experimental results for additional target models, further comparison between IDP-SGD and DPSGD, and detailed per-target model statistics are presented in Figure 9, Figure 10 and Table 12 in Appendix D.3. The results highlight that the individual privacy assignment of our IDP-SGD has a practical impact and indeed protects data points with different levels of privacy to different degrees. Additionally, the privacy risk for data points with $\varepsilon = 10$ when training with IDP-SGD is lower than when training purely on them with standard DP-SGD. This privacy gain does not come at large expense of points with $\varepsilon = 20$ whose privacy risk remains roughly the same(see Figure 10).

## 5   Comparison to Other Methods

**Comparison to Individualized PATE.**   We compare against Individualized PATE [6] (IPATE), the only other work aiming at enabling individual privacy assignment in ML. See details of their method and the general PATE framework in Appendix B.3. For the comparison, we report the student model accuracies of IPATE for the MNIST, SVHN, and CIFAR10 datasets. The results are presented in Table 13 in Appendix D.4. We observe that IDP-SGD constantly outperforms IPATE with the same privacy budget assignment.

**Synergy between Individualized Privacy Accounting and Assignment.**   We show that we can leverage synergies between our individualized privacy assignment and the orthogonal line of work on privacy accounting (see Section 2.2). Therefore, we extend the approach by Feldman and Zrnic [11] with our method of assigning each data point its individual privacy budget through an individualized

Table 4: **Comparison between standard DP-SGD, individual accounting, and individual accounting+assignment.** We present the test accuracy (%) on MNIST for training with $\varepsilon = 0.3$ for standard DP-SGD and individual accounting [11], while setting $\varepsilon = 0.30$ and $\varepsilon = 0.31$ for half of the points each when using individual accounting and assignment. Combining individualized privacy accounting and assignment further increases utility.

| VANILLA DP-SGD | INDIVIDUAL ACCOUNTING | INDIVIDUAL ACCOUNTING AND ASSIGNMENT |
|---|---|---|
| $93.29 \pm 0.49$ | $93.64 \pm 0.46$ | $94.16 \pm 0.23$ |

Renyi-filter. This filter causes the point to be excluded from training once its individually assigned privacy budget is exhausted. Note that in contrast, in the approach by Feldman and Zrnic [11], all data points obtain the same privacy budget and data points are excluded from training once they reach this budget. For more details, see the description in Appendix D.5.

To assess the performance of pure individual accounting, we assign the same privacy budget of $0.3$ to all $60,000$ training data points in MNIST, following the experimental setup by Feldman and Zrnic [11]. We then enable the individual privacy assignment and change the privacy values to $\varepsilon = 0.3$ for the first half of the points and $\varepsilon = 0.31$ for the second half of data points. We empirically observe that in this low-$\varepsilon$ regime, the small change of $\varepsilon$ for half the data points is enough to cause significant improvement in the accuracy of the final model, as we show in Table 4. Note that the differences in accuracy reported for standard DP-SGD on MNIST between Table 4 and Table 8 differ. This is because of the different privacy budgets ($\varepsilon = 0.3$ vs. $\varepsilon = 1.0$). Additionally, in Figure 3, we present the number of active data points, *i.e.*, data points that did not yet exhaust their individual privacy budget, over training. We find that many data points are used longer during training with individualized assignment compared to using only individualized accounting. This is because the data points with a higher privacy budget exhaust their budget later and, thus, can be used longer. We are only able to run this experiment on MNIST as done in Feldman and Zrnic [11] since the method is based on full-batch gradient descent: gradients are computed over the whole dataset at once, which limits its applicability to large and high-dimensional datasets.

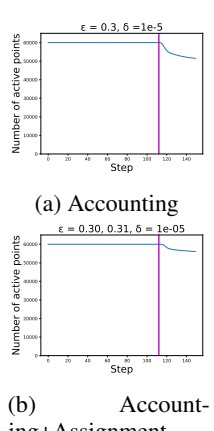

(a) Accounting

(b) Accounting+Assignment

Figure 3: **Number of Active Data Points Over Training.**

## 6 Conclusion and Future Work

Prior work on ML with DP guarantees assigns a uniform privacy budget $\varepsilon$ over the entire dataset. This approach fails to capture that different individuals have different expectations towards privacy and also decreases the model's ability to learn from the data. To overcome the limitations of a uniform privacy budget and to implement individual users' privacy preferences, we propose two modifications to the popular DP-SGD algorithm. Our Sample and Scale mechanisms adapt training to yield individual per-data point privacy guarantees and boost utility of the trained models. For future work, we believe that our individualized privacy assignment should be closely integrated with a form of practical individual privacy accounting. This could enable us to obtain a fine-grained notion of individualized privacy guarantees that tightly meet the users' expectations.

## Acknowledgements

We would like to acknowledge our sponsors, who support our research with financial and in-kind contributions: this includes CIFAR through the Canada CIFAR AI Chair and NSERC through the Discovery Grant. Resources used in preparing this research were provided, in part, by the Province of Ontario, the Government of Canada through CIFAR, and companies sponsoring the Vector Institute. As team member of the Center for Trustworthy Artificial Intelligence (CTAI), Christopher Mühl was funded by the German Federal Ministry for the Environment, Nature Conservation, Nuclear Safety and Consumer Protection. We would also like to thank CleverHans lab group members for their feedback.

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

# A  Broader Impacts and Ethical Considerations of this Work

The specification of an adequate privacy level $\varepsilon$ is challenging since it does not only depend on the domain but also on the dataset itself. This challenge persists when assigning individual privacy levels over a dataset. Especially in sensitive domains, one needs to make sure that the assignment of individual privacy guarantees does not pose an additional risk to the individuals whose data is processed. Therefore, one needs to first make sure that assigning individual privacy guarantees is not abused. This can occur when the entity training the model violates underlying individuals' privacy by nudging them into choosing inadequate privacy levels. Second, one needs to prevent individuals from giving up their privacy due to a lack of understanding or a wrong perception of their risk. To do so, IDP-SGD should not be used as a stand-alone method. Instead, following the example of [6], we suggest incorporating it into a whole process that involves communicating the functioning of DP and the risks associated with a decision to the individuals, analyzing their preferences, and supporting them in an informed decision-making process. There exist a growing body of work on the communication of DP methodology and associated risks to the public [41, 9, 12]. The most recent finding [12] suggests that medical risk communication formats, such as percentages or frequencies, are best applicable to inform about privacy risks. For the identification of privacy preferences, existing frameworks, such as [37] can be applied. Finally, we suggest giving individuals the choice between categorical privacy levels (*e.g.*, *high*, *medium*, *low*) while putting a regulatory entity, such as an ethics committee in charge of specifying concrete numeric values of $\varepsilon$. This mitigates the disadvantages of the general limited interpretability of $\varepsilon$ that exist independently of individualization.

## A.1  Limitations

In this work, we consider a setup where individuals indicate their privacy preferences. We acknowledge that not all individuals are aware of their own privacy preferences and might, therefore, provide inaccurate indications. Therefore, our framework should be always deployed in a protected setup as described above. While we provide theoretical privacy guarantees in the framework of differential privacy, we assess the practical impact of these guarantees on the individuals solely through membership inference attacks. Future work should investigate the practical (disparate per privacy-group) impact on other known privacy risks, such as data reconstruction. Due to the lack of sensitive-real world datasets that, in addition to individuals' data, also capture their privacy preferences, we evaluate our methods on standard benchmark datasets. We, therefore, have to *simulate* a privacy preference distribution on this data according to the distribution known from society [17, 5].

## A.2  Confidentiality of the Privacy Budgets

In some setups, an individual's choice of privacy budget could reflect some sensitive information. Imagine a medical context with a dataset that consists of individuals that do and individuals that do not have cancer. The latter ones might be more inclined towards choosing higher privacy protection to hide their condition. If an attacker with access to the trained model was able to deduce the individual's privacy budget, they might, in turn, be able to draw conclusions on the individual's medical state.

Yet, theoretical results from ML, *e.g.*, [14], have shown that one cannot currently perform sample-efficient black-box audits to determine the privacy-budget of a trained model. Frameworks to estimate privacy of a trained model, such as [39] rely on training 1000-100,000 shadow models to get an estimate of the full model's privacy guarantee. In our IDP-SGD setup, the model does not have one single $\varepsilon$, but one $\varepsilon_p$ per group, aggravating the impracticability of auditing guarantees and limiting the applicability of existing frameworks. As a result, the adversary cannot currently determine the different per-group privacy budgets of the model.

Our MIA experiments in Section 4.3 show a difference in privacy risks over entire privacy groups. First note that these experiments (Figure 2) were the most costly ones in our paper. We had to train more than 1500 (3 times 512) shadow models to generate the results. Yet, there are two limitations: a) The results only show differences in the groups, but do not reveal the respective $\varepsilon_p$ values, b) they do not allow to perform per-point distinctions. This is because the distributions of privacy risks over the two groups still have a large overlap. Hence, given a single point's risks and the two distributions, one cannot predict without error to what group the point belongs. This error could be reduced when the privacy budgets between the groups differ significantly (0.1 vs. 100). Note however, that we suggest deploying the model in a setup where individuals simply specify their privacy preferences

and the model builder or an ethics committee assign concrete epsilon values to these preferences. We observed that extreme differences between privacy groups' are less beneficial because they degrade overall model performance. Hence, in practice, these large differences are very unlikely. As a result, it will be very challenging for an adversary to determine which privacy group a data point belongs to. Even if they manage to, they are unlikely to be able to learn the privacy budget of the group as described above.

To enhance confidentiality of privacy budgets, Alaggan et al. [2] implement a form of deniability for users' membership in a certain privacy group by uniformly sampling the privacy values over groups in an overlapping way: the unconcerned sample from 1., the pragmatists from 0.5, 0.75, 1., and the fundamentalists from 0, 0.5, 1, such that on average, the values will be 1., 0.75, and 0.5 within the groups (0 denotes the strongest, while 1 is the weakest privacy guarantee). Such an approach can be implemented into our IDP-SGD framework as well. The drawback is that an individual who expresses strong privacy preferences (fundamentalists) might actually be assigned a privacy value 1 by randomness, which causes the same privacy leakage as for an unconcerned individual. Depending on the individuals' preferences and the application, when protecting privacy group membership is more important than protecting the actual data, this approach might still be valid and this approach could be integrated with IDP-SGD, as well.

## B  Extended Background

### B.1  Differential Privacy

Differential Privacy (DP) [10] provides a theoretical upper bound on the influence that a single data point can have on the outcome of an analysis over the whole dataset.

The most commonly applied instantiation of DP is $(\varepsilon, \delta)$-DP which is defined as follows:

**Definition B.1.** Let $D, D' \subseteq \mathcal{D}$ be two neighboring datasets, *i.e.*, datasets that differ in exactly one data point. Let further $M: \mathcal{D}^* \to \mathcal{R}$ be a mechanism that processes an arbitrary number of data points. $M$ satisfies $(\varepsilon, \delta)$-DP if for all datasets $D \sim D'$, and for all result events $R \subseteq \mathcal{R}$

$$\mathbb{P}\left[M(D) \in R\right] \leq e^{\varepsilon} \cdot \mathbb{P}\left[M(D') \in R\right] + \delta . \tag{4}$$

In the definition, the privacy level is specified by $\varepsilon \in \mathbb{R}_+$, while $\delta \in [0, 1]$ offers a relaxation, *i.e.*, a small probability of violating the guarantees.

### B.2  Differential Privacy Accounting in Machine Learning

Privacy accounting in DP-SGD is most commonly implemented by the moments accountant which keeps track of a bound on the moments of the privacy loss random variable at outcome $R$, defined by

$$c(R; M, \mathsf{aux}, D, D') = \log \frac{\Pr[M(\mathsf{aux}, D) \in R]}{\Pr[M(\mathsf{aux}, D') \in R]}. \tag{5}$$

for $D, D', M$ and $R$ as above, and an auxiliary input $\mathsf{aux}$.

### B.3  PATE Algorithm

The Private Aggregation of Teacher Ensembles (PATE) algorithm [33] represents an alternative to DP-SGD for training ML models with privacy guarantees. This ensemble-based algorithm implements privacy guarantees through a knowledge transfer from the ensemble to a separate student model. More concretely, in PATE, the private training data is split into non-overlapping subsets and distributed among several teacher models. Once each teacher is trained on their own data subset, they perform a privacy-preserving knowledge transfer by jointly labeling an additional unlabeled public dataset. To implement DP guarantees, noise is added to the labeling process. On completion, an independent student model is trained on the public dataset using the generated labels, and thereby incorporating knowledge about the original training data without ever requiring access to it.

### B.4  Individualized PATE.

The individualized PATE variants by [6] are *Upsample* and *Weight*. Upsample duplicates data points and distributes them to different teachers in the PATE ensemble. Utility increase in this method

result from the availability of more training data points for the teachers. Since the sensitivity for a duplicated data point increases (it can change the votes of all the teachers that are trained on its duplicates), privacy consumption of that data point is higher. Since each data point can be duplicated individually, this method allows for a fine-grained individualization. In contrast, the weight method allows for a per-teacher model privacy individualization. Data points with the same privacy budget are assigned to the same teacher model(s) and the impact of that teacher model's weight on the final vote is weighted according to its training data points' privacy preferences. Teachers that are trained on data points with low privacy requirements are weighted higher, increasing the privacy consumption of their training data.

### B.5   The LiRA Membership Inference Attack

The LiRA membership inference attack [8] proceeds in three steps to determine which data points from a dataset $\mathcal{D} = \{x_1, \ldots, x_N\}$ were used to train a target model $f$: (1) First, multiple shadow models, similar to $f$, are trained on different subsets of $\mathcal{D}$. (2) Then, the mean and variance of two loss distributions $\mathcal{N}(\mu_{\text{in}}, \sigma_{\text{in}})$ and $\mathcal{N}(\mu_{\text{out}}, \sigma_{\text{out}})$ are estimated per data point $x_i$. Both distributions are calculated from the logits of $x_i$ at the target class $y_i$—the former one over the shadow models that $x_i$ is a member of, the latter one on shadow models that $x_i$ is not a member of. (3) Finally, the likelihood of a new data point $x$ of class $y$ being a member of the target model $f$ is calculated as $\Lambda = \frac{p(f(x)_y \mid \mathcal{N}(\mu_{\text{in}}, \sigma_{\text{in}}^2))}{p(f(x)_y \mid \mathcal{N}(\mu_{\text{out}}, \sigma_{\text{out}}^2))}$.

## C   Details on the Methods

### C.1   Sample

**Leveraging Higher Sampling Rates for Increased Utility.**   With higher sampling rates for certain data points, utility could, in principle be increased in several ways. (1) Larger sampling rates can be used to obtain higher mini-batch sizes $B$ (while keeping the number of training iterations $I$ constant). Line 3 in Algorithm 1 shows that noise is added to the aggregate of all gradients. Hence, with larger mini-batches, the signal-to-noise ratio is higher, which can improve training. (2) Alternatively, the mini-batch size $B$ can be kept constant while increasing the number of training iterations $I$. Longer training can increase model performance. However, these two approaches result in a change of core training hyperparamters (mini-batch size and number of iterations). As we discuss in Section 3, changing training hyperparameters would require a separate fine-tuning, for example, to adapt the learning rate for larger mini-batches, as in (1) or longer training as in (2). Since the training parameters would change according to the privacy budgets encountered in the private training dataset, and the ratios of these budgets over the training data points, the hyperparameter-tuning would have to be repeated whenever the dataset is updated, individuals change their privacy preferences, or decide to withdraw their consent for leveraging their data for the ML model alltogether, yielding significant overheads. We, therefore, implement our Sample according to the third option (3), described in Section 3.3 which leverages higher sampling rates for improved utility by reducing the noise multiplier of the added noise $\sigma$. This allows us to perform an apple to apple comparison between both our methods and to the standard DP-SGD.

### C.2   Scale

**Deriving Noise Multiplier $\sigma_{\text{scale}}$.**   Given the desired clip norm $c$ found through hyperparameter tuning of the standard DP-SGD, we set the individual clip norms such that their weighted average yields $c$ as $c = \frac{1}{N} \sum_{p=1}^{P} |G_p| \cdot c_p$. So, we derive the $\sigma_{\text{scale}}$ in the following way:

$$c = \frac{1}{N} \sum_{p=1}^{P} |G_p| \cdot c_p \tag{6}$$

$$c = \frac{1}{N} \sum_{p=1}^{P} |G_p| \cdot \left( c \frac{\sigma_{\text{scale}}}{\sigma_p} \right) \tag{7}$$

$$c = c \sigma_{\text{scale}} \frac{1}{N} \sum_{p=1}^{P} \frac{|G_p|}{\sigma_p} \tag{8}$$

$$\sigma_{\text{scale}} = \left( \frac{1}{N} \sum_{p=1}^{P} \frac{|G_p|}{\sigma_p} \right)^{-1} \tag{9}$$

From (6) to (7), we use the equality between the scale of added noise $\sigma_{\text{scale}} c = \sigma_p c_p$. In (8), we extract terms that are independent of the privacy groups ($c$ and $\sigma_{\text{scale}}$) before the summation.

### C.3  Algorithmic Details

We specify our used sub-routines used for determining a sample rate or noise multiplier based on given privacy parameters in Algorithm 4 and Algorithm 5, respectively.

---

**Algorithm 4: Subroutine getSampleRate.**  Is the equivalent to Opacus' function get_noise_multiplier [32] for deriving an adequate sample rate for given paramteers.

---

**Require:** Target $\varepsilon$, target $\delta$, iterations $I$, noise multiplier $\sigma$, precision $\gamma = 0.01$
1: **init** $\varepsilon_{\text{high}}$: $\varepsilon_{\text{high}} \leftarrow \infty$
2: **init** $q_{\text{low}}, q_{\text{high}}$: $q_{\text{low}} \leftarrow 1\mathrm{e}{-9}, q_{\text{high}} \leftarrow 0.1$
3: **while** $\varepsilon_{\text{high}} > \varepsilon$ **do**
4:     $q_{\text{high}} \leftarrow 2 q_{\text{high}}$
5:     $\varepsilon_{\text{high}} \leftarrow I \cdot 2 q_{\text{high}}^2 \frac{1}{\sigma^2}$ {approximate $\varepsilon$ according to Equation (2), we suppress $\alpha$ for simplicity}
6: **end while**
7: **while** $\varepsilon - \varepsilon_{\text{high}} > \gamma$ **do**
8:     $q \leftarrow (q_{\text{low}} + q_{\text{high}})/2$
9:     $\varepsilon_{\text{temp}} \leftarrow I \cdot 2 q^2 \frac{1}{\sigma^2}$ {approximate $\varepsilon$ according to Equation (2), we suppress $\alpha$ for simplicity}
10:     **if** $\varepsilon_{\text{temp}} < \varepsilon$ **then**
11:         $q_{\text{high}} \leftarrow q$
12:         $\varepsilon_{\text{high}} \leftarrow \varepsilon_{\text{temp}}$
13:     **else**
14:         $q_{\text{low}} \leftarrow q$
15:     **end if**
16: **end while**
17: **Output** $q_{\text{high}}$

---

## D  Additional Empirical Evaluation

We report the hyperparameters found for our individualized methods in Table 6. The training and standard DP-SGD hyperparameters are specified in Table 5. The performance of our individualized methods when using the hyperparameters of standard DP-SGD is presented in Table 8. Already when using these (non-individually tuned) hyperparameters, our methods yield a significant performance increase in comparison to standard DP-SGD. For MNIST, individual hyperparameter for our methods and individual setups did not yield significant improvements, therefore the results presented in Table 8 and Table 2 are identical for MNIST.

**Computing Resources.**  The implementation of our methods does not increase computation time over the standard implementation of DP-SGD apart from the derivation of the privacy parameters that is performed once at the beginning of training. Hence, to run all experiments around our methods and

**Algorithm 5: Subroutine getNoise.** Implements Opacus' function get_noise_multiplier [32].

---

**Require:** Target $\varepsilon$, target $\delta$, iterations $I$, sample rate $q$, precision $\gamma = 0.01$
 1: **init** $\varepsilon_{\text{high}}$: $\varepsilon_{\text{high}} \leftarrow \infty$
 2: **init** $\sigma_{\text{low}}, \sigma_{\text{high}}$: $\sigma_{\text{low}} \leftarrow 0, \sigma_{\text{high}} \leftarrow 10$
 3: **while** $\varepsilon_{\text{high}} > \varepsilon$ **do**
 4:     $\sigma_{\text{high}} \leftarrow 2\sigma_{\text{high}}$
 5:     $\varepsilon_{\text{high}} \leftarrow I \cdot 2q^2 \frac{1}{\sigma_{\text{high}}^2}$ {approximate $\varepsilon$ according to Equation (2), we suppress $\alpha$ for simplicity}
 6: **end while**
 7: **while** $\varepsilon - \varepsilon_{\text{high}} > \gamma$ **do**
 8:     $\sigma \leftarrow (\sigma_{\text{low}} + \sigma_{\text{high}})/2$
 9:     $\varepsilon_{\text{temp}} \leftarrow I \cdot 2q^2 \frac{1}{\sigma^2}$ {approximate $\varepsilon$ according to Equation (2), we suppress $\alpha$ for simplicity}
10:     **if** $\varepsilon_{\text{temp}} < \varepsilon$ **then**
11:         $\sigma_{\text{high}} \leftarrow \sigma$
12:         $\varepsilon_{\text{high}} \leftarrow \varepsilon_{\text{temp}}$
13:     **else**
14:         $\sigma_{\text{low}} \leftarrow \sigma$
15:     **end if**
16: **end while**
17: **Output** $\sigma_{\text{high}}$

---

their evaluation, we required, in total less than 16h of GPU time on a standard GeForce RTX 2080 Ti. We ran the experiment on combining individualized privacy assignment and accounting on the same machines RTX 2080Ti and the total compute time is also around 2h. To train all the shadow models for our membership inference attack and run inference on them, we ran on an A100 GPU and required a total runtime of roughly 32 hours.

Table 5: **DP-SGD Hyperparameters.** LR: learning rate, B: expected mini-batch size, I: number of iterations, C: clip norm, $\sigma$: noise multiplier in DP-SGD derived from the desired privacy budget $\varepsilon = 1$. Default target $\delta = 0.00001$.

| DATASET | LR | B | I | C | $\sigma$ |
|---|---|---|---|---|---|
| MNIST | 0.6 | 512 | 9375∼80 EPOCHS | 0.2 | 3.42529 |
| SVHN | 0.2 | 1024 | 2146∼30 EPOCHS | 0.9 | 2.74658 |
| CIFAR10 | 0.7 | 1024 | 1465∼30 EPOCHS | 0.4 | 3.29346 |

Table 6: **DP-SGD Hyperparameters (Individually Tuned).** LR: learning rate, B: expected mini-batch size, I: number of iterations, C: clip norm, $\sigma$. Default target $\delta = 0.00001$. Setup A is for privacy budgets $\varepsilon = \{1.0, 2.0, 3.0\}$ and their respective distribution of *34%-43%-23%*. Setup B is for the same privacy budgets but with their distributions *54%-37%-9%*.

| DATASET | METHOD | SETUP | LR | B | I | C | $\sigma$ |
|---|---|---|---|---|---|---|---|
| MNIST | SAMPLE | A | 0.6 | 512 | 9375∼80 EPOCHS | 0.2 | 3.42529 |
| SVHN | SAMPLE | A | 0.2 | 1024 | 5723∼80 EPOCHS | 0.6 | 2.53261 |
| CIFAR10 | SAMPLE | A | 0.2 | 1024 | 2929∼60 EPOCHS | 1.0 | 2.65712 |
| MNIST | SAMPLE | B | 0.6 | 512 | 9375∼80 EPOCHS | 0.2 | 3.42529 |
| SVHN | SAMPLE | B | 0.1 | 1024 | 3577∼50 EPOCHS | 0.6 | 2.41421 |
| CIFAR10 | SAMPLE | B | 0.1 | 1024 | 2929∼60 EPOCHS | 1.8 | 3.14049 |
| MNIST | SCALE | A | 0.6 | 512 | 9375∼80 EPOCHS | 0.2 | 3.42529 |
| SVHN | SCALE | A | 0.1 | 1024 | 3577∼50 EPOCHS | 2.0 | 2.09719 |
| CIFAR10 | SCALE | A | 0.2 | 1024 | 3418∼70 EPOCHS | 1.1 | 2.88335 |
| MNIST | SCALE | B | 0.6 | 512 | 9375∼80 EPOCHS | 0.2 | 3.42529 |
| SVHN | SCALE | B | 0.1 | 1024 | 3577∼50 EPOCHS | 1.6 | 2.45703 |
| CIFAR10 | SCALE | B | 0.1 | 1024 | 2929∼60 EPOCHS | 1.8 | 3.17792 |

We present the individualized privacy parameters identified for our methods in Table 7.

Table 7: **Individualization Parameters Computed by our Methods for Table 8.** We report the individualized privacy parameters identified for our Scale and Sample by Algorithm 3 and Algorithm 2, respectively. The parameters are obtained on the MNIST, SVHN, and CIFAR10 datasets when using the privacy budget distributions of Table 8 with $\varepsilon = \{1.0, 2.0, 3.0\}$

| DATASET | SETUP | DP-SGD | | | SCALE | | | SAMPLE | |
| --- | --- | --- | --- | --- | --- | --- | --- | --- | --- |
| | | $\sigma$ | $c$ | $q$ | $\sigma_{\text{SCALE}}$ | $\{\sigma_1, \ldots, \sigma_P\}$ | $\{c_1, \ldots, c_P\}$ | $\sigma_{\text{SAMPLE}}$ | $\{q_1, \ldots, q_P\}$ |
| MNIST | *34%-43%-23%* | 3.425 | 0.2 | 0.008 | 2.063 | {2.189, 1.310, 1.032} | {0.129, 0.216, 0.274} | 2.024 | {0.005, 0.009, 0.013} |
| | *54%-37%-9%* | 3.425 | 0.2 | 0.008 | 2.418 | {2.189, 1.310, 1.032} | {0.148, 0.248, 0.315} | 2.376 | {0.006, 0.011, 0.016} |
| SVHN | *34%-43%-23%* | 2.747 | 0.9 | 0.014 | 1.896 | {2.747, 1.589, 1.214} | {0.561, 0.970, 1.270} | 1.667 | {0.008, 0.015, 0.021} |
| | *54%-37%-9%* | 2.747 | 0.9 | 0.014 | 2.180 | {2.747, 1.589, 1.214} | {0.651, 1.125, 1.472} | 1.937 | {0.009, 0.018, 0.025} |
| CIFAR10 | *34%-43%-23%* | 3.293 | 0.4 | 0.020 | 2.244 | {3.294, 1.868, 1.399} | {0.244, 0.430, 0.574} | 1.965 | {0.012, 0.022, 0.031} |
| | *54%-37%-9%* | 3.293 | 0.4 | 0.020 | 2.594 | {3.294, 1.868, 1.399} | {0.285, 0.502, 0.671} | 2.300 | {0.014, 0.026, 0.037} |

Table 8: **Model Test Accuracy after training with Standard DP-SGD vs our Individualized DP-SGD** using Sample or Scale. **D** is the distribution of privacy groups (percentages) and $\varepsilon$ the privacy budget for a given group. The percentages of the three privacy groups are chosen according to Alaggan et al. [2] (first setup) and [30] (second setup). We used the hyperparameters found for standard DP-SGD, see Table 5 and report the standard deviation over 10 trials.

| DATASET | | SETUP | DP-SGD | SAMPLE | SCALE |
| --- | --- | --- | --- | --- | --- |
| MNIST | **D** | *34%-43%-23%* | 96.75 | **97.81** | 97.78 |
| | $\varepsilon$ | *1.0-2.0-3.0* | ± 0.15 | **± 0.09** | ± 0.08 |
| | **D** | *54%-37%-9%* | 96.75 | **97.6** | 97.54 |
| | $\varepsilon$ | *1.0-2.0-3.0* | ± 0.15 | **± 0.11** | 0.09 |
| SVHN | **D** | *34%-43%-23%* | 83.26 | **84.56** | 84.48 |
| | $\varepsilon$ | *1.0-2.0-3.0* | ±0.31 | **±0.25** | ±0.25 |
| | **D** | *54%-37%-9%* | 83.26 | **84.32** | 84.31 |
| | $\varepsilon$ | *1.0-2.0-3.0* | ±0.31 | **±0.31** | ±0.26 |
| CIFAR10 | **D** | *34%-43%-23%* | 52.77 | 54.89 | **54.92** |
| | $\varepsilon$ | *1.0-2.0-3.0* | ± 0.65 | ± 0.55 | **±0.63** |
| | **D** | *54%-37%-9%* | 52.77 | 54.88 | **55.00** |
| | $\varepsilon$ | *1.0-2.0-3.0* | ± 0.65 | ± 0.45 | **± 0.65** |

### D.1 Privacy Consumption of our Methods

We track privacy consumption of our methods over the course of training in Figure 4. The figure highlights the good calibration of our methods which causes all privacy groups to exhaust their budget after the pre-specified number of training iterations.

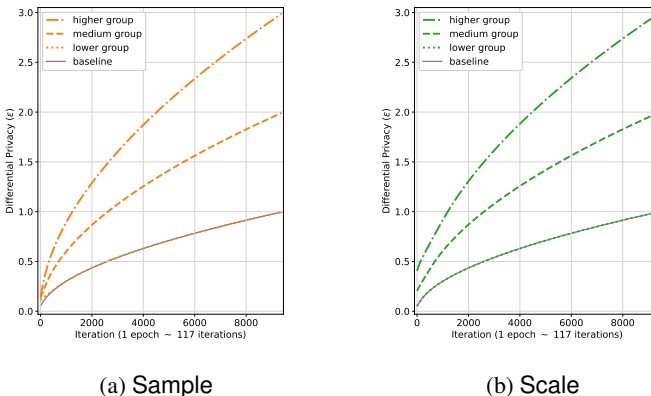

(a) Sample  (b) Scale

Figure 4: **Individual Privacy Costs on MNIST** for $\varepsilon \in \{1, 2, 3\}$ with Distribution $(54\%, 37\%, 9\%)$.

### D.2 General Applicability of our Methods

We showcase the practical impact of individual privacy assignments on individual utility and demonstrate how our methods extend to many privacy groups and privacy budget distributions.

Table 9: **100 privacy groups on MNIST.** We randomly assign MNIST training data points to one of the 100 privacy groups with respective $\varepsilon_p$ sampled uniformly at random from $[1, 6]$. We compare against the DP-SGD baseline with $\varepsilon = 1$ and observe that our methods converge well and outperform the baseline.

|  | DP-SGD | Sample | Scale |
|---|---|---|---|
| Test Acc. (%) | 96.75±0.15 | 98.17±0.18 | 98.21±0.1 |

**Practical Impact.** We run experiments on the CIFAR10 dataset where we assign higher or lower privacy budgets to one of the 10 classes. We select all data points from the class 0 (which we can think of, for example, as an underprivileged group) as one privacy group and assign to it either higher ($\varepsilon = 3$), the same ($\varepsilon = 2$), or lower ($\varepsilon = 1$) privacy budgets in comparison to all other data points from the other classes ($\varepsilon = 2$). Table 10 shows that the choice of privacy budget for a single group also impacts the other groups. We observe that by only changing the privacy budget for the selected group (in this case for class 0), we can flip its performance (its accuracy from being higher to being lower) in comparison to the accuracy of the other group (consisting of remaining classes). In the example of class 0, the accuracy is 67.76% when assigned high privacy budget ($\varepsilon = 3$), which is a higher accuracy than for all other classes that have an average accuracy of around 53.41% and assigned the privacy budget $\varepsilon = 2$. Then, by modifying only the privacy budget of class 0 and by assigning to it the low privacy budget ($\varepsilon = 1$), its accuracy drops to a mere 25.76% and is below the accuracy of 56.58% for the remaining classes. We visualize the impact of the chosen privacy budget on utility over all classes (instead of only the class 0) in Figure 5 and Figure 6 for CIFAR10 and MNIST, respectively.

**More Privacy Groups.** We present in Table 11 the test accuracy for ten privacy groups, corresponding to the ten classes of the CIFAR10 dataset when each of the privacy groups obtains a different privacy budget. We obtained these budgets by manually tuning them such that the accuracy gap between the privacy groups is minimized. We also visualize the accuracy over training in Figure 7. The figure visualizes that our methods are able to make all privacy groups converge to similar accuracies.

To evaluate IDP-SGD in the limit, we perform additional experiments with 100 privacy groups by randomly assigning MNIST training data points to one of the 100 privacy groups with respective $\varepsilon_p$ sampled uniformly at random from $[1, 6]$. Our results in Table 9 indicate that IDP-SGD is able to handle this large number of privacy groups without any degradation in terms of convergence: both our methods, Sample and Scale, continuously outperform the standard DP-SGD baseline with $\varepsilon = 1$ for all data points.

### D.3 Additional Results for MIA

Membership inference success for a single target model of our Sample and Scale methods is shown in Figure 8. In Figure 9, we present the results over for 5 different target models for Sample. The figure highlights that over all target models, the two privacy groups' privacy risk is different: the group with higher protection $\varepsilon = 10$ constantly has a lower AUC than the group with lower protection $\varepsilon = 20$. The test statistics over the different privacy groups' LiRA likelihood scores for all five target models are shown in Table 12.

In Figure 10, we also compare the MIA risk of our IDP-SGD with the risk of training with DP-SGD, using the LiRA attack. The results show that the privacy risk for data points with $\varepsilon = 10$ when training with IDP-SGD is lower than when training with standard DP-SGD. This privacy gain does not come at large expense of points with $\varepsilon = 20$ whose privacy risk remains roughly the same.

### D.4 Comparison to Individualized Privacy with IPATE

We present the comparison between our IDP-SGD and IPATE [6] in Table 13. In PATE, accuracy refers to the student model accuracy. The results in Table 13 are averaged over three experiments for IPATE and ten runs for IDP-SGD. Note that the accuracies we report for IPATE differ from the accuracy values reported by [6] in Table 1, since they report average voting accuracy (*i.e.*, how correct are individual teacher model votes), whereas we report the resulting student model accuracies, which

Table 10: **Accuracy for Subgroups.** We assess the accuracy of subgroups when their privacy budgets differ. We select a single class for a given group and assign either higher, the same, or lower privacy budgets in comparison to groups with other classes. We change the privacy budgets only for bolded classes in a given experiment while all other classes have the same privacy budget ($\varepsilon = 2$).

| Classes | Privacy Budget | | |
|---|---|---|---|
| | Higher ($\varepsilon = 3$) | Same ($\varepsilon = 2$) | Lower ($\varepsilon = 1$) |
| **0** | $\mathbf{67.76 \pm 2.14}$ | $55.24 \pm 1.98$ | $25.76 \pm 2.52$ |
| 1-9 | $53.41 \pm 2.2$ | $54.72 \pm 2.49$ | $\mathbf{56.58 \pm 2.29}$ |
| **1** | $\mathbf{80.84 \pm 1.2}$ | $72.79 \pm 1.6$ | $46.09 \pm 3.91$ |
| 0,2-9 | $51.65 \pm 2.28$ | $52.77 \pm 2.53$ | $\mathbf{54.59 \pm 2.23}$ |
| **2** | $51.31 \pm 3.53$ | $36.59 \pm 3.33$ | $9.53 \pm 1.41$ |
| 0-1,3-9 | $\mathbf{54.26 \pm 2.84}$ | $56.79 \pm 2.34$ | $\mathbf{58.87 \pm 2.18}$ |
| **3** | $52.88 \pm 2.6$ | $32.64 \pm 1.91$ | $6.67 \pm 1.09$ |
| 0-2,4-9 | $\mathbf{54.62 \pm 2.42}$ | $57.23 \pm 2.5$ | $\mathbf{58.75 \pm 2.42}$ |
| **4** | $\mathbf{56.99 \pm 1.87}$ | $40.06 \pm 2.88$ | $9.44 \pm 1.68$ |
| 0-3,5-9 | $54.41 \pm 2.21$ | $56.41 \pm 2.39$ | $\mathbf{58.18 \pm 2.02}$ |
| **5** | $\mathbf{64.11 \pm 2.27}$ | $51.86 \pm 2.21$ | $15.04 \pm 2.31$ |
| 0-4,6-9 | $53.28 \pm 2.16$ | $55.1 \pm 2.46$ | $\mathbf{57.54 \pm 2.49}$ |
| **6** | $\mathbf{73.54 \pm 2.36}$ | $65.8 \pm 4.25$ | $40.6 \pm 4.18$ |
| 0-5,7-9 | $52.05 \pm 2.21$ | $53.55 \pm 2.23$ | $\mathbf{56.06 \pm 2.33}$ |
| **7** | $\mathbf{68.22 \pm 1.17}$ | $61.15 \pm 1.99$ | $41.08 \pm 2.75$ |
| 0-6,8-9 | $52.5 \pm 2.43$ | $54.07 \pm 2.49$ | $\mathbf{56.01 \pm 2.76}$ |
| **8** | $\mathbf{77.39 \pm 1.24}$ | $68.53 \pm 2.34$ | $37.42 \pm 3.37$ |
| 0-7,9 | $51.51 \pm 2.54$ | $53.25 \pm 2.45$ | $\mathbf{55.28 \pm 2.37}$ |
| **9** | $\mathbf{72.82 \pm 1.48}$ | $63.08 \pm 1.9$ | $32.79 \pm 2.73$ |
| 1-8 | $52.05 \pm 2.32$ | $53.85 \pm 2.5$ | $\mathbf{56.06 \pm 2.52}$ |

corresponds to the final performance of the method. Note that IPATE does not apply the performance improvements suggested by Papernot et al. [34] (*e.g.*,virtual adversarial training) or MixMatch which could increase the student model's performance.

### D.5    Integrating Privacy Accounting and Assignment

The main goal behind individualized accounting is to obtain tighter privacy analysis (recall our discussion in Section 2.2). Instead of tracking a single privacy loss estimate across all data points, an individual privacy loss is kept track of for each data point. Feldman and Zrnic [11] defines a new individual privacy filter, which drops data points that exceed the individualized privacy loss from further processing. However, the same privacy budget is assigned to each data point. The individual assignment of a privacy budget to each data point is a natural extension of individualized accounting and can be directly incorporated into this framework. Therefore, a given data point has its own privacy filter and is dropped from the analysis once the filter indicates that the data point's privacy budget is exhausted.

### D.6    Combining **Sample** and **Scale**

We run additional experiments to showcase the advantage of combining both our methods. The combination of **Sample** and **Scale** extends the number of hyperparameters and offers the potential to find better performing algorithms. It can be implemented as follows: obtain individual sampling probabilities as in **Sample**. Once a mini-batch is sampled, use the respective data points' indices and

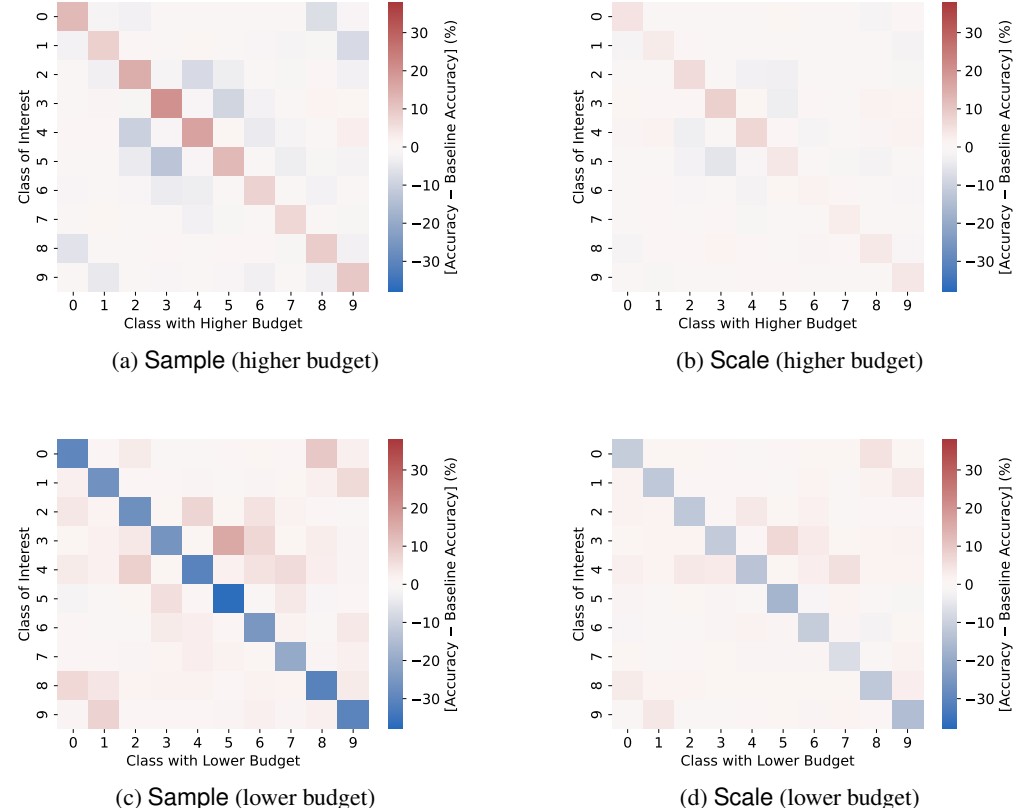

(a) Sample (higher budget)

(b) Scale (higher budget)

(c) Sample (lower budget)

(d) Scale (lower budget)

Figure 5: **CIFAR10: Accuracy Changes for Subgroups.** We assess how the test **Accuracy** of a **Class of interest** changes in comparison to the **Baseline Accuracy** (standard DP-SGD with $\varepsilon = 2$) when we, during training, assign a lower ($\varepsilon = 1$) or a higher ($\varepsilon = 3$) privacy budget to data points from a class (shown on the x-axis). The diagonals show that by increasing a class' privacy budget (lower privacy), their utility increases, while it decreases with the decrease of privacy budget (higher privacy). Similar results for MNIST can be found in Figure 6.

clip their gradients as in Scale. We allow to weight the different methods to arbitrary fractions (e.g., 50% Sample, 50% Scale), and derive the privacy parameters such that all privacy groups exhaust their respective budgets at the end of the specified number of iterations. For practical evaluation, we use the MNIST dataset and the first privacy setup (0.34, 0.43, 0.23) with privacy budgets $\varepsilon = 1, 2, 3$ and different weightings as indicated in Table 14.

The baseline (first and last rows) are equivalent to Table 2 in the main paper, while the additional three rows in between with the new results represent average test accuracy and the standard deviation over ten independent random runs over different weights assigned to the respective method.

Our results demonstrate a clear progression from the lower performing Scale approach to the better performing Sample. This aligns with expectations based on our implementation, where we combine the methods based on the parameters found for them sequentially.

Note that each of our methods has its advantages and disadvantages and the combination of both methods provides more options to balance (dis)advantages. For example, when some points' sample rates are extremely small, then the model might never see those points during training and the combined method could reduce this possibility while still improving over the Scale-only method.

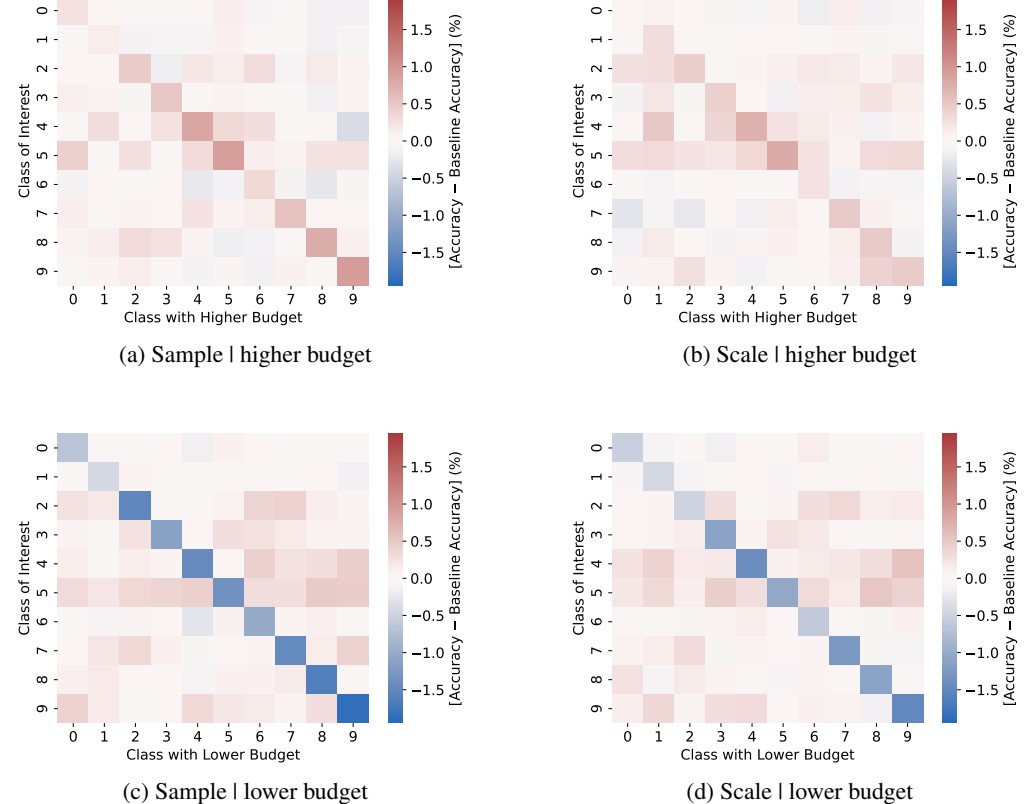

(a) Sample | higher budget

(b) Scale | higher budget

(c) Sample | lower budget

(d) Scale | lower budget

Figure 6: **MNIST: Accuracy Changes for Subgroups.** We assess how the test accuracy of a class changes (in comparison to standard DP-SGD with $\varepsilon = 2$) when we, during training, assign a lower ($\varepsilon = 1$) or a higher ($\varepsilon = 3$) privacy budget to data points from this class. The diagonals show that by increasing a class' privacy budget (lower privacy), their utility increases, while it decreases with the decrease of privacy budget (higher privacy).

### D.7 Examining Convergence of IDP-SGD

Individualization directly affects the objective function of the training procedure. To verify that our individualized methods maintain a smooth convergence, we examine the loss history of one group of training data in the MNIST dataset and compare it with non-individualized DP-SGD.

We collected the loss values (and accuracy values) over the 4 following setups on the MNIST dataset and privacy setup 2 ($54\%, 37\%, 9\%, \varepsilon = \{1, 2, 3\}$):

1. loss value for privacy group with $\varepsilon = 1$ for the Sample method;
2. loss value for privacy group with $\varepsilon = 1$ for the Scale method;
3. training with standard DP-SGD ($\varepsilon = 1$) solely on the $54\%$ of data points that have the privacy budget of $\varepsilon = 1$, and
4. training with the standard DP-SGD ($\varepsilon = 1$) on $100\%$ of the MNIST dataset.

In the following table, we depict the training loss of the privacy group with $\varepsilon = 1$ during training:

| epochs | Sample | Scale | DP-SGD ($54\%$ data) | DP-SGD ($100\%$ data) |
|---|---|---|---|---|
| 20 | **0.145**±0.010 | 0.148±0.011 | 0.211±0.018 | 0.162±0.013 |
| 40 | **0.116**±0.005 | 0.121±0.006 | 0.187±0.014 | 0.143±0.007 |
| 60 | **0.109**±0.005 | 0.115±0.006 | 0.183±0.010 | 0.142±0.008 |
| 80 | **0.105**±0.005 | 0.112±0.005 | 0.184±0.008 | 0.142±0.006 |

Table 11: **Per-class Individual Privacy Assignments.** We manually optimize the per-class individual privacy budgets for Sample such that the model achieves the same accuracy over all classes. The resulting per-class privacy budgets yield the maximum gap $\Delta$ between the highest and lowest accuracy level of only 0.39% for Sample, and 0.88% for Scale. For the baseline ($\varepsilon = 3$ for all classes) $\Delta = 2.03$ is significantly higher, highlighting that our approach can successfully minimize the accuracy gap between different privacy groups. We run the experiment on the MNIST dataset and report average per-class test-accuracies over three separate runs. See each privacy group's test accuracy over training in Figure 7.

| Class | 0 | 1 | 2 | 3 | 4 | 5 | 6 | 7 | 8 | 9 | $\Delta$ |
|---|---|---|---|---|---|---|---|---|---|---|---|
| Baseline ($\varepsilon = 3$) | 98.95 | 99.06 | 98.39 | 98.09 | 97.93 | 98.47 | 98.16 | 98.12 | 97.78 | 97.03 | 2.03 |
| Budgets | 0.75 | 0.5 | 2.0 | 2.6 | 4.1 | 2.1 | 2.05 | 3.0 | 3.1 | 6.1 | / |
| Sample | 98.16 | 98.09 | 98.16 | 97.95 | 98.10 | 97.91 | 97.77 | 97.99 | 98.02 | 97.89 | 0.39 |
| Scale | 98.44 | 98.36 | 98.13 | 98.02 | 97.76 | 98.17 | 97.91 | 97.96 | 97.91 | 97.56 | 0.88 |

Table 12: **Statistical differences between Privacy Groups.** We conduct a student t-test to determine if the Lira likelihood scores for data points with privacy budget $\varepsilon = 10$ differ from the ones of data points with $\varepsilon = 20$. All results with $p < 0.05$ indicate statistically significant differences. Results for Sample.

| Target Model | $\Delta$ | $p$ |
|---|---|---|
| 1 | 5.16 | 2.49e-07 |
| 2 | 2.41 | 0.016 |
| 3 | 1.84 | 0.066 |
| 4 | 4.03 | 5.52e-05 |
| 5 | 2.537 | 0.011 |

The loss values correspond to the following accuracy values:

| epochs | Sample | Scale | DP-SGD (54% data) | DP-SGD (100% data) |
|---|---|---|---|---|
| 20 | **96.114**±0.215 | 96.024±0.257 | 94.071±0.460 | 95.692±0.315 |
| 40 | **97.024**±0.095 | 96.906±0.102 | 95.078±0.333 | 96.358±0.191 |
| 60 | **97.296**±0.108 | 97.123±0.126 | 95.306±0.228 | 96.525±0.169 |
| 80 | **97.430**±0.106 | 97.257±0.115 | 95.362±0.187 | 96.560±0.125 |

Our results indicate that the availability of more information from the data of different privacy groups in our Sample and Scale methods enables the model to learn better features. This leads to lower loss for the data points with privacy budget $\varepsilon = 1$ than if the model was trained on only those 54% of points. The effect of including more data (i.e., using 100% of the MNIST dataset), all with $\varepsilon = 1$, is weaker, indicating that even the data with highest privacy requirement benefits substantially from our individualized privacy in terms of reduced loss and increased accuracy.

### D.8 Runtime Analyses

We analyze the runtime of our methods in comparison to standard DP-SGD. The runtime is determined by two steps, the first one being the calculation of the individualized parameters of the methods which is executed exactly once before the training. The second one is the actual training time. For the calculation of individualized parameters, Table 16 indicates that the runtime depends on the number of privacy groups $P$ specified for the implementation of Algorithm 2 and Algorithm 3 and the precision $\gamma$ (see Algorithm 5). We observe a linear growth in runtime w.r.t. the number of privacy groups. Note that, as stated above, this additional runtime occurs only once with IDP-SGD, namely before training starts. Also note that when we repeat experiments with the same privacy setups, the parameters do not need to be derived every time, but can be set directly based on one initial computation. In terms of actual training time, Table 16 indicates that Sample and Scale do not

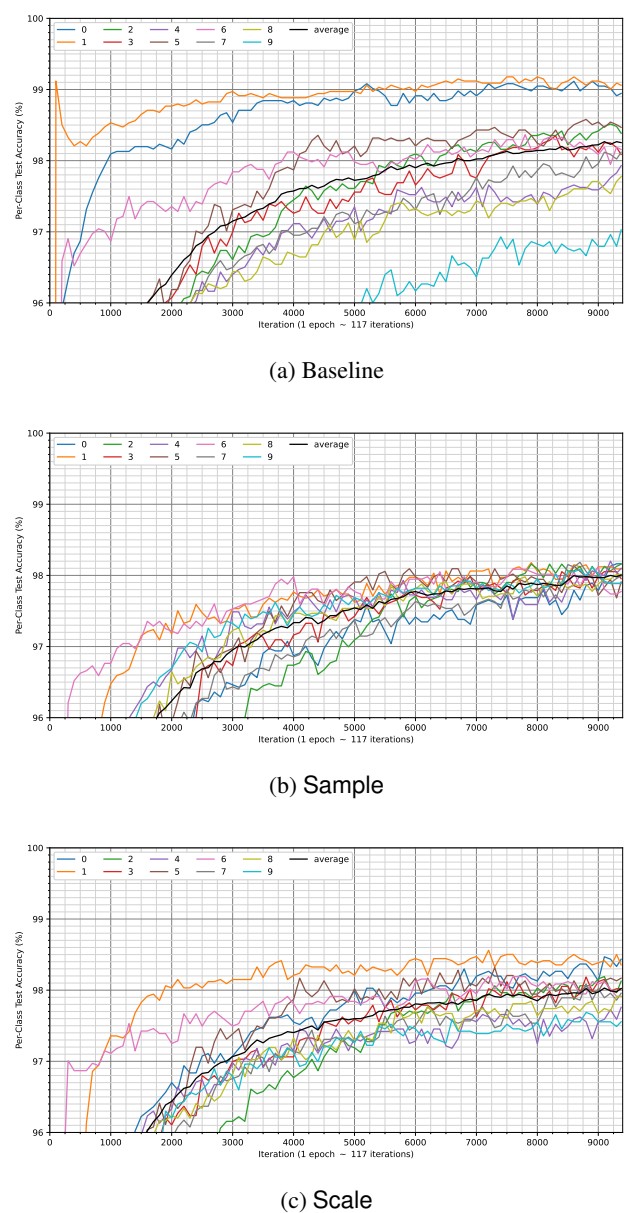

(a) Baseline

(b) Sample

(c) Scale

Figure 7: **Per-class test accuracies over CIFAR10 training with Per-Class Privacy Budgets.** We manually tune the per-class privacy budgets for Sample to obtain the same per-class accuracy at the end of training, see Table 11. Comparison with the Baseline (a) where all classes obtain $\varepsilon = 3$ highlight that Sample (b) and Scale (c) successfully reduce the accuracy gap between the different classes.

significantly increase the training time in comparison to standard DP-SGD which is mainly due to their tight integration with the standard DP-SGD algorithm and the opacus library.

# E    Alternative Baselines

Throughout this work, we compare our methods with Standard DP-SGD which cannot take into account different privacy requirements at the same time. Thus, we consider it to apply the highest privacy protection required to all data points equally. Nonetheless, we can think of two more ways to

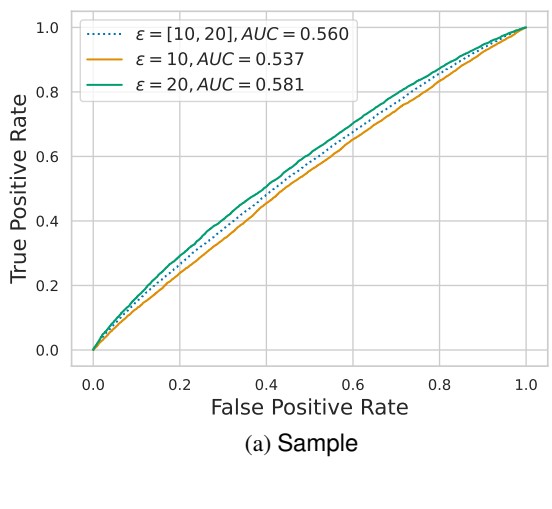

(a) Sample

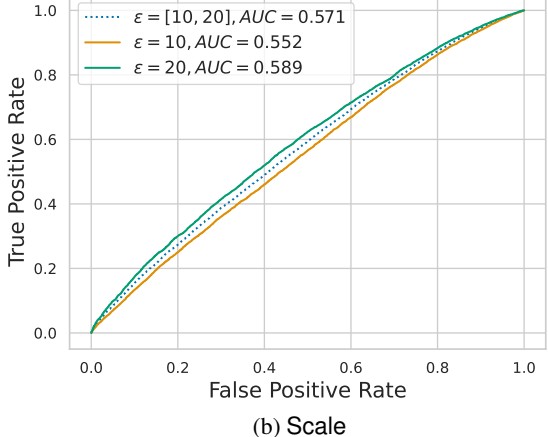

(b) Scale

Figure 8: **True-Positive rate vs. False-Positive Rate of Lira Membership Inference Attacks Per Privacy-Budget.** We follow the same setup as in Figure 2. We show the single target model for both (a) Sample and (b) Scale methods.

use Standard DP-SGD on data having heterogeneous privacy requirements. Those approaches have different benefits and drawbacks and might perform better than our chosen baseline approach in some scenarios.

### E.1 Exclude Lower Privacy Groups

Instead of applying the strongest privacy protection, the deciding ML expert could entirely exclude data of low privacy groups from training for loosening the restrictions on the remaining data points' influence on model updates. In some cases, it would be worth giving up the information and privacy budgets of those lower privacy groups to achieve utility improvements. This approach performs poorly if important information is wasted, *e.g.*, most data of one class has the highest privacy requirement.

### E.2 Learn Privacy Groups Separately

It is also possible to make use of all privacy budgets, independent of their diversity, although only using Standard DP-SGD. Namely, a model can be trained on each privacy group separately one after another, whereby the corresponding lowest budget is regarded for each group. A drawback of this approach is that the model could forget its knowledge about previously learned privacy groups.

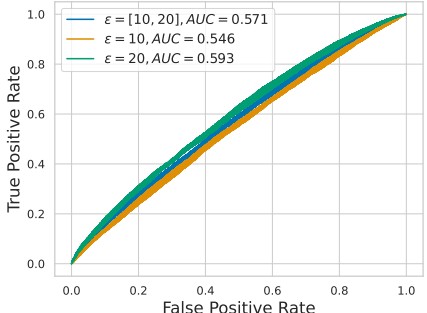

Figure 9: **True-Positive rate vs. False-Positive Rate of Lira Membership Inference Attacks Per Privacy-Budget.** We follow the same setup as in Figure 2. We run the experiment for five different target models and aggregate the results with the error bars for the Sample method.

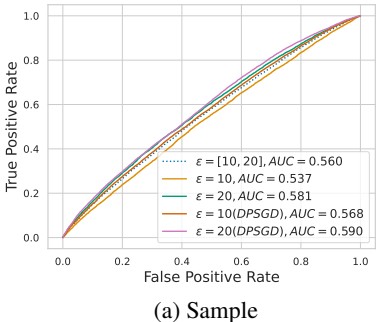

(a) Sample

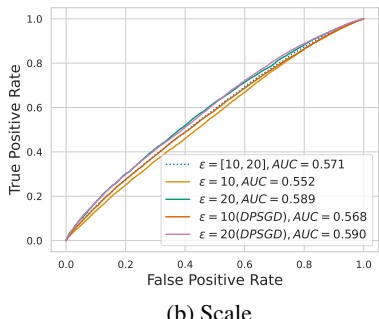

(b) Scale

Figure 10: **MIA on IDP-SGD vs. MIA on standard DP-SGD.** In addition to the first three lines (blue, orange, and green) from Figure 9, we also added the red and pink lines. The original three lines refer to a model trained with IDP-SGD on two privacy groups (12,500 data points with $\varepsilon = 10$ and 12,500 with $\varepsilon = 20$). For the red line, we trained 512 shadow models with standard DP-SGD and 25,000 CIFAR10 data points that all have $\varepsilon = 10$. For the pink line, we did the same with 25,000 data points and $\varepsilon = 20$. Then, we performed MIA according to the LiRA attack over the two setups.

### E.3 Empirical Comparison of Baselines

We empirically evaluate against these two additional baselines using the MNIST dataset. For baseline E.1, we include all data points with a privacy budget of $\varepsilon \geq 2$ und use $\varepsilon = 2$ as the privacy budget for training. After hyperparameter tuning, the training on the remaining data points (43%+23% and 37%+9% of the total data) yields the accuracy reported in Table 17.

For baseline E.2, we also did hyperparameter tuning and used the best noise multiplier of 2.5 for training. We trained the groups sequentially, always continuing training with the next group once the privacy budget of the previous groups was exhausted. We evaluated both starting with the privacy group that has loosest and strongest preferences (orders [3,2,1] and [1,2,3], respectively). Starting with the group that has strongest privacy requirements and ending on the group that has loosest privacy requirements yielded the best results which we report in Table 17.

In summary, we observe that our methods outperform the other baselines.

## F Alternative Individualization

We present the alternative ways of individualizing privacy guarantees in DP-SGD that we considered in the design process of IDP-SGD and describe their drawbacks.

Table 13: **Comparison between IDP-SGD and Individualized PATE (IPATE).** We select the weighting mechanism from IPATE that performs better than the upsampling method. The *Setup* indicates the size (in %) of privacy groups. We present the accuracy (%) for training with $\varepsilon = \{1.0, 2.0, 3.0\}$ privacy budgets, respectively to the order of the privacy groups.

| DATASET | SETUP | BASELINE PATE | IPATE | SAMPLE | SCALE |
|---------|-------|---------------|-------|--------|-------|
| MNIST | 34%-43%-23% | 91.17± 1.25 | 95.27 ± 0.33 | **97.81** | 97.78 |
|  | 54%-37%-9% |  | 95.74 ± 0.43 | **97.6** | 97.54 |
| SVHN | 34%-43%-23% | 22.46 ± 5.19 | 41.45 ± 1.69 | **84.56** | 84.48 |
|  | 54%-37%-9% |  | 44.64 ± 0.55 | **84.32** | 84.31 |
| CIFAR10 | 34%-43%-23% | 24.83 ± 1.56 | 33.20 ± 0.94 | 54.89 | **54.92** |
|  | 54%-37%-9% |  | 35.59 ± 0.73 | 54.88 | **55.00** |

Table 14: **Combining Sample and Scale.** We combine Sample and Scale according to the weights indicated in the table and perform experiments on the MNIST dataset. The respective percentages highlight to what degree which method contributed. For example, 0% Sample and 100% Scale means that only Scale was executed (like in the results of the main paper in Table 2), while 50% Sample and 50% Scale expresses that both methods contributed equally. The results are averaged over ten independent runs.

| Sample (weight in %) | Scale (weight in %) | Test Accuracy (in %) |
|:---:|:---:|:---:|
| 0 | 100 | 97.78±0.08 |
| 25 | 75 | 97.75±0.10 |
| 50 | 50 | 97.80±0.10 |
| 75 | 25 | 97.80±0.10 |
| 100 | 0 | 97.81±0.09 |

## F.1 Individual Per-Data Point Noise

Individualized privacy could, in principle also be obtained by adding different amounts of noise to different data points. Every of the $P$ privacy group would have their individual $\{\sigma_1, \ldots, \sigma_P\}$. Utility improvements would result from some data points requiring smaller amounts of added noise. Note however, that in DP-SGD, while clipping is performed on a per-data point basis, noise addition is performed on a per-mini-batch basis (line 8 in Algorithm 1). Hence, there are two possibilities to implement individual noise addition: either (i) by operating on mini-batch sizes of 1, or (ii) by implementing a two-step sampling approach which first randomly samples a privacy group for a given training iteration and then applies the standard Poisson sampling to obtain the mini-batch consisting of data points from this group. While both approaches are conceptually correct, they exhibit significant drawbacks. Approach (i) first slows down training performance due to more operations requiring to be carried out on individual data points, rather than a mini-batch. Second, due to the weak signal-to-noise ratio when adding noise to individual gradients, model performance is likely to degrade. Finally, sampling cannot be performed with Poisson anymore since with Poisson sampling, it is not possible to pre-determine and specify exact mini-batch sizes, instead these depend on the outcome of the random sampling process. Approach (ii) could overcome the first two issues. However, the different groups sizes are still strictly smaller than the entire dataset and large parts of DP-SGD's degrading the tight privacy bounds obtained by privacy amplification through subsampling.[8] The privacy amplification through subsampling allows to scale down the noise $\sigma$ by the factor $B/N$ (with $B$ being the expected mini-batch size, $N$ the total number of data points, and $B \ll N$) while still ensuring the same $\varepsilon$ as with $\sigma$ [20]. This privacy amplification is crucial to the practical performance (privacy-utility trade-offs) of DP-SGD. Hence, by using the $P$ privacy groups of sizes $\{N_1, \ldots, N_P\}$ with $N_i \ll N$, the factors $B/N_i \ll B/N$ for all $i\{1, \ldots, P\}$. This effect cancels out, or in the worst

---

[8]Note that there exist other, less popular approaches to implement DP in ML than the DP-SGD algorithm, such as Differentially Private Follow-the-Regularized-Leader (DP-FTRL) which do not rely on subsampling but instead obtain tighter privacy bounds from adding correlated noise over the training iterations. However, since DP-FTRL under-performs DP-SGD for high-privacy regimes, and unfolds it advantages mainly in FL scenarios where the same data is only learned from once, or with a small number of epochs, we consider the approach outside of the scope of this work.

Table 15: **Runtime in seconds to obtain privacy parameters (clip norms or sample rates) for different numbers of groups $P$ and precisions $\gamma$.** We report the time it takes for our methods to generate the privacy parameters for the first setup of Table 2 of the original submission (34%-43%-23%).

| | $\gamma = 0.01$ | | $\gamma = 0.0001$ | |
|-----|--------|-------|--------|-------|
| $P$ | Sample | Scale | Sample | Scale |
| 2 | 3.24 | 0.57 | 5.79 | 1.01 |
| 8 | 10.34 | 2.51 | 20.22 | 4.78 |
| 32 | 34.2 | 10.24 | 70.89 | 19.75 |
| 128 | 127.78 | 35.13 | 277.52 | 78.85 |

Table 16: **Runtime in seconds for our methods.** We report runtime when training on the CIFAR10 dataset and the CNN architecture for the first privacy setup of Table 2 in the original submission (34%-43%-23%) with $\varepsilon = 1, 2, 3$. Experiments are conducted on an A100-GPU. Runtimes are averaged over three trials and average with standard deviation is reported.

| DP-SGD | Sample | Scale |
|--------|--------|-------|
| $403.67 \pm 2.31$ | $406.33 \pm 0.58$ | $414.33 \pm 3.21$ |

Table 17: **Empirical evaluation against other baselines.** We report the obtained test accuracy obtained with our two methods vs. two other baselines for individualized privacy on the MNIST dataset. Similar to Table 8, we use $\varepsilon = 1, 2, 3$. Both our Sample and Scale outperform the other baselines.

| Setup | DP-SGD | E.1 Baseline | E.2 Baseline | Sample | Scale |
|-------|--------|--------------|--------------|--------|-------|
| 34%-43%-23% | 96.75 | 97.6 | 97.4 | 97.81 | 97.78 |
| 54%-37%-9% | 96.75 | 97.1 | 97.3 | 97.6 | 97.54 |

case even inverts the privacy-utility benefits that should arise from assigning individual data points less noise based on their privacy preference in our individualization.

### F.2 Duplicating Data Points

When duplicating data points in the training dataset, similar to the Upsampling mechanism in IPATE [6], the DP-SGD algorithm itself does not need to be adapted. Instead, different privacy levels of different data points stem from their individual number of replication within the training data. This approach offers a very fine-grained control on individual privacy levels, since, in principle, each data point could be replicated a different number of times. Utility gain would result from the larger training dataset. However, this type of upsampling opens the possibility for the same data point to be present multiple times in the mini-batch used for training in a given iteration. This stands in contrast to the original DP-SGD, where participation of each data point for training at a given iteration is determined by an independent Bernoulli trial, and hence, a data point can be either included once or not at all in a mini-batch. The possibility for a data point to be included multiple times $n$ inside the same mini-batch changes the sensitivity of the mechanism from $c$ to $nc$. According to [27], when noise is added according to $\sigma$, a mechanism with sensitivity $nc$ is $(\alpha, \frac{2(nc)^2}{2\sigma^2})$-RDP. The quadratic influence of the sensitivity to privacy bound results in a severe increase in the RDP $\varepsilon$, making the approach suboptimal in terms of privacy-utility guarantees. Additionally, upsampling leads to an effective increase in a data point's the sample-rate which further increases privacy costs.

## G  Additional Proofs

### G.1  Additional Proofs for Individualized Privacy

**Proof for Lemma 3.1**

*Proof.* First note that $(\varepsilon_1, \delta)$-DP can be considered as a special case of $(\{\varepsilon_1, \varepsilon_2, \ldots, \varepsilon_P\}, \delta)$-IDP, where $\forall p \in [1, P]\ \varepsilon_p = \varepsilon_1$. We can, hence apply Equation (3) and see that an $M$ that satisfies $(\varepsilon_1, \delta)$-DP has a privacy guarantee of $\mathbb{P}[M(D) \in R] \leq e^{\varepsilon_1} \cdot \mathbb{P}[M(D') \in R] + \delta$. Given that by our definition $\forall p \in [2, P]$ it holds that $\varepsilon_p > \varepsilon_1$, for all $p$, it holds that $\mathbb{P}[M(D) \in R] \leq e^{\varepsilon_1} \cdot \mathbb{P}[M(D') \in R] \leq e^{\varepsilon_p} \cdot \mathbb{P}[M(D') \in R] + \delta$. From this inequality, it follows that $M$ also satisfies $(\{\varepsilon_1, \varepsilon_2, \ldots, \varepsilon_P\}, \delta)$-IDP. $\square$

**Proof for Lemma 3.2**

*Proof.* Analogous to the previous proof, by our definition, it holds that $\forall p \in [1, P-1]$ the $\varepsilon_p < \varepsilon_P$. From an $M$ that satisfies $(\{\varepsilon_1, \varepsilon_2, \ldots, \varepsilon_P\}, \delta)$-IDP, it, therefore, holds that for all $p$ the $\mathbb{P}[M(D) \in R] \leq e^{\varepsilon_p} \cdot \mathbb{P}[M(D') \in R] + \delta \leq e^{\varepsilon_P} \cdot \mathbb{P}[M(D') \in R] + \delta$. This inequality shows that $M$ satisfies $(\varepsilon_P, \delta)$-DP. $\square$

## G.2 Privacy Proofs for our Methods

Either of our methods can be considered as an individual SGM with different parameters from the point of view of each privacy group. This is because for each group, we have to examine neighboring datasets which differ in an arbitrary data point from that group. Our Sample method ensures an individual sample rate for all points of each group, while our Scale method applies an individual noise multiplier for all points of each group.

So, assume we have $P$ different privacy groups. We base our proofs on the privacy guarantees of Sample and Scale on the observation that our methods can be considered as $P$ simultaneously executed Subsampled Gaussian Mechanisms (SGM) [28] that update the same model. We first note that every data point's privacy preference stays stable over the course of training, therefore, conceptually, the privacy groups divide our training data in $P$ disjoint subsets. Hence, each of the $P$ SGMs operates on a different partition of the training data, using different individual sample rates or clip norms. One can think of the simultaneous model updates by multiple SGMs with different privacy parameters as a similar concept to federated learning [26] with different local privacy guarantees [40]: each client holds a disjoint subset of data, executes some local model update, implements privacy guarantees through local subsampling of the data, clipping, and noise addition (potentially with different parameters than other clients), and then shares the model updates with a server that aggregates all updates and applies them to the shared model.

Within each training step, our methods' SGMs also have a different privacy consumption due to their different sample rates and clip norms (sensitivity). Given that we keep the intermediate model states (checkpoints) private and only release the final model [42], the privacy costs in RDP over multiple training iterations add up linearly per SGM, such that, at the end each SGM yields $(\alpha, \bar{\varepsilon}_p)$-RDP, $p \in [1, \ldots, P]$. As a consequence, the algorithm over all privacy groups yields $(\alpha, \{\bar{\varepsilon}_1, \bar{\varepsilon}_2, \ldots, \bar{\varepsilon}_P\})$-RDP which can then be converted to $(\{\varepsilon_1, \varepsilon_2, \ldots, \varepsilon_P\}, \delta)$-DP using standard conversion [27] as we will show in the following.

**Theorem G.1.** *Our Sample mechanism satisfies $(\{\varepsilon_1, \varepsilon_2, \ldots, \varepsilon_P\}, \delta)$-DP.*

*Proof.* We prove the bound for any particular privacy group separately. Fix $p \in \{1, \ldots, P\}$, let $D \subseteq \mathcal{D}$ be the training dataset, and select any $x_i \in D$ that belongs to group $\mathcal{G}_p$. We are interested in comparing outcomes of mechanism $M$ on $D$ with its outcomes on $D' = D \setminus \{x_i\}$ where $M$ represents a particular model update of Sample. We get Gaussian mixtures

$$M(D') = \sum_{L \subset D} \pi_L \mathcal{N}\left(f(L), \sigma_{\text{sample}}^2 \mathbf{I}^d\right) \quad \text{and}$$

$$M(D) = \sum_{L \subset D} \pi_L \left((1 - q_p) \mathcal{N}\left(f(L), \sigma_{\text{sample}}^2 \mathbf{I}^d\right) + q_p \mathcal{N}\left(f(L \cup \{x_i\}), \sigma_{\text{sample}}^2 \mathbf{I}^d\right)\right) \quad,$$

where $f(L)$ is the clipped gradient of the current mini-batch $L$, $\pi_L$ is its probability, $\sigma_{\text{sample}} > 0$ is the noise scale, $\mathbf{I}^d$ is the identity matrix, and $0 < q_p \leq 1$ is the individual sample rate of $x_i$ and every other point in $\mathcal{G}_p$. Note that $\sigma_{\text{sample}}$ is actually multiplied by the clip norm $c_{\text{sample}} > 0$. Gaussian mechanisms are invariant regarding scaling of their sensitivity and noise scale, but instead depend on the relationship between sensitivity and noise scale, called the noise multiplier. Hence, we can ignore $c_{\text{sample}}$ and consider $f$ to have sensitivity 1.

Now we can see that the Gaussian mixtures of our **Sample** are equivalent to those corresponding to the original SGM from Mironov et al. [28], Thm. 4, when we parameterize it with sample rate $q_p$ and noise scale $\sigma_{\text{sample}}$ which are individual per group. Therefore, all RDP bounds of the original SGD apply, especially $(\alpha, \bar{\varepsilon}_p)$-RDP with $\bar{\varepsilon}_p = 2q_p^2 \frac{\alpha}{\sigma_{\text{sample}}^2}$ in a particular parameter regime (cfg. Thm. 11 from Mironov et al. [28]). As a final step of the proof, we need to convert from $(\alpha, \bar{\varepsilon}_p)$-RDP guarantees to $(\varepsilon_p, \delta)$-DP guarantees following Mironov [27] (see Section 2.1). Note that our individual parameters have been selected before the start of training so that each group's privacy budget is exhausted at the intended number of iterations (see Algorithm 2). □

**Theorem G.2.** *Our **Scale** mechanism satisfies* $(\{\varepsilon_1, \varepsilon_2, \ldots, \varepsilon_P\}, \delta)$*-DP.*

*Proof.* This proof is analog to the proof of Theorem G.1 with the difference that we have a global sample rate $q$ but individual noise multipliers $\sigma_p$ and clip norms $c_p$. Moreover, Algorithm 3 is used to configure parameters prior to training. □

