# OpenReview forum: "Have it your way: Individualized Privacy Assignment for DP-SGD"
_NeurIPS.cc/2023/Conference — NeurIPS 2023 poster_

### Official Review · Reviewer_r49G · 2023-06-17

**Soundness:** 3 good
**Presentation:** 4 excellent
**Contribution:** 3 good
**Rating:** 6
**Confidence:** 4

**Summary:**

This paper designs variants of differentially private SGD to satisfy different privacy expectations, e.g., users can choose one from high, medium, or low levels of privacy. There are two variants of DP-SGD, one changes the sampling probability of different groups, and the other changes the clipping threshold of different groups.

**Strengths:**

1. The Sample and Scale methods are easy-to-implement. Their design aligns nicely with the fact that DP-SGD adds noise to the aggregated gradient, instead of separate noise for individual gradients.

2. Experiments on several common vision datasets show that individualized DP-SGD is better than DP-SGD with a uniform (the smallest) privacy parameter. The authors also run membership inference attacks to show that groups with smaller privacy parameters do have better empirical privacy.

3. The paper is very well written and easy to follow.


**Weaknesses:**


1. The privacy expectations in the current version are independent of model accuracy. In practice, the minority groups, whose accuracy is usually worse, may prefer stronger privacy than other groups. What would happen if the group with the worst accuracy has the strongest/weakest privacy expectation? One way to implement this is to extend the experiments in Appendix D.2 by assigning a larger/smaller privacy parameter to the class with the worst non-private accuracy.

2. The privacy expectation itself may be data dependent and hence the individualized privacy assignment cannot be made public, as the authors discussed in Line 68 – Line 73. For example, in a medical dataset, users diagnosed with certain diseases may have higher privacy expectations.

3. The authors use a shallow convolutional network for the experiments. It would be better to see whether the results still hold for deep learning models, e.g., ResNets or transformers.

4. The authors only report the average accuracy. It would be better if the authors could show the accuracy of different groups. A recent line of work shows that DP has disparate impact on model accuracy [1]. Will the groups with smaller privacy assignments lose more accuracy?

[1]: Differential Privacy Has Disparate Impact on Model Accuracy. https://arxiv.org/abs/1905.12101


**Questions:**

See Weaknesses

**Limitations:**

---

> ### Author Rebuttal · Authors · 2023-08-09
>
> We thank the reviewer for the comments and provide our responses below.
>
> > **W1: Influence on utility for underprivileged groups**
>
> We already present such a result in Table 8  in our submission. We selected the CIFAR10 dataset and assigned lower or higher privacy budget to one of the classes. Class 3 can be considered as an underprivileged group since it has the lowest accuracy across all classes of only 32.64%. If the privacy budget for this class is decreased further to eps=1 while the other classes maintain eps=2, the accuracy for class 3 plummets to only 6.67%, while the accuracy for other classes increases. In contrast, when a higher privacy budget of eps=3 is assigned to class 3, then its accuracy increases substantially and is comparable with accuracy for the remaining classes.
>
> > **W2: Confidentiality of the privacy values**
>
> There is no need to reveal the privacy assignments and the end user of the released model should be oblivious to the training data points’ individual privacy preferences. In the general response, we detail how the fact that end users can only interact with the trained model preserves confidentiality of the training data points’ privacy preferences.
>
> > **W3: Extending experimentation to ResNet and Transformers**
>
> We thank the reviewer for their suggestion and we extended our experimentation. Among other experiments (see general response), we
> - trained a ResNet18 from scratch with CIFAR10
> - and fine-tuned a BERT transformer (bert-base-uncased) on the SNLI dataset for natural language inference (NLI).
>
> A summary of the results is summarizes in this table for convenience:
> | Model    | Dataset | DP-SGD (eps=5) | Sample (eps=5,7.5,10) | Scale (eps=5,7.5,10) |
> |----------|---------|----------------|--------------------|-------------------|
> | ResNet18 | CIFAR10 | 47.52+-0.84    | 48.52+-0.69        | 48.77+-0.73       |
> | BERT     | SNLI    | 75.91+-0.23    | 76.11+-0.21        | 76.5+-0.17        |
>
> In both experiments, our results show that the Scale and Sample methods again outperform standard DP-SGD. See the general response for more experimental details.
>
> > **W4: Per privacy-subgroup utility**
>
> We show the accuracy for different subgroups with their different privacy budgets in Table 8. We consider each subgroup to be a different class. We consider the test data to divide into the same per-class subgroups as the training data. This enables us to show a per-group (class) test accuracy under the given different privacy budgets. The observed trend indicates that if a group is willing to sacrifice more privacy, it will gain higher utility of the model whereas for groups with stronger privacy preferences, the opposite is the case.
>
> We would also like to clarify that in the main paper, for example Table 2, we, again report the test accuracies. Test data points do not get the possibility to specify any privacy budget (since the model only predicts on them and does not train on them, their privacy is not leaked anyways). For the training data points in the main paper, we assigned their privacy budget at random during training. Hence, there is no mapping to what subgroup of the test data would correspond to what subgroup of the training data. As a result, when reporting test accuracies in this setup, it is impossible to show per-group test accuracies.

---

> > ### Comment · Reviewer_r49G · 2023-08-14
> >
> > Thank you for your response. Regards W1, it may be better to clarify that Class 0 is the underprivileged group in Appendix D.2.
> >
> > After reading the response and reviews from other reviewers, I decided to maintain my positive score.

---

> > > ### Author Response · Authors · 2023-08-15
> > > **Thank you for the feedback**
> > >
> > > We thank the reviewer for the positive feedback and maintaining the score. We will extend the description in Appendix D.2 and include the term "the underprivileged group".

---

### Official Review · Reviewer_Sruy · 2023-06-30

**Soundness:** 3 good
**Presentation:** 2 fair
**Contribution:** 3 good
**Rating:** 5
**Confidence:** 4

**Summary:**

This paper proposes extensions of the DP-SGD algorithm to support individualized differential privacy (called the IDP-SGD approach). Unlike traditional differential privacy, which imposes a single privacy budget epsilon to all data points, the data points may now have different privacy budgets. Two extensions are proposed. First, the Sample method adjust the sampling rates of data points depending on their privacy budgets. Second, the Scale method adjusts the individual clip norms to effectively add individualized noise. The noise itself cannot be directly individualized due to the DP-SGD algorithm behavior of adding per-batch noise. Experiments show that IDP-SGD outperforms DP-SGD and IPATE in terms of utility and can be complemented with privacy accounting techniques.

**Strengths:**

* Proposes novel extensions of DP-SGD to support individualized privacy.
* The Sample and Scale methods are easy to deploy and have theoretical guarantees.
* Experiments show that IDP-SGD outperforms baselines (DP-SGD and I-PATE) and can be used with privacy accounting.

**Weaknesses:**

W1) The core contributions of the paper are not contained in the main part of the paper.
  * The theoretical analysis should be more specific because the proposed individual privacy notion with the extended $\delta$ term is different from previous personalized DP and conventional DP. In the proof of Theorem G.1., the original SGM seems to be about conventional DP and RDP, but the author simply applied it to individual privacy without specific proof about its properties.

  * The paper shows the algorithm that finds parameters, but it does not provide a way to adapt the parameter to the original DP-SGD. Because DP-SGD uses a single sample rate and a single clip norm, adapting multiple sampling rate and clipping norm is also a new mechanism. It seems that a more specific explanation is needed in terms of implementation. For example, the original DP-SGD assumes Poisson sampling in principle, but approximates it due to computational cost. However, this is possible because the sampling probability is fixed, and it seems necessary to explain how to transform this to satisfy independent sampling.

  * It is difficult to know the exact process of the algorithm. Algorithm 3 is a very important part of the paper, but it is difficult to understand properly because it is omitted in the main part.

W2) It is not clear how theoretically optimal Sample and Scale are. Looking at Table 1, $\epsilon_p$ of Sample and Scale could have a loose bound depending on the $\sigma_{\mathrm{SAMPLE}}$ and $\sigma_p$ values, respectively. It would be helpful to actually see what these values are in the experiments.

W3) In Algorithm 1, the paper says that getSampleRate is equivalent to the Opacus function get_noise_multiplier without a detailed description, but the two functions seem to return different objects. The function getSampleRate returns a sample rate, but the Opacus function get_noise_multiplier returns noise_mutiplier $\sigma$.

W4) The comparison between Sample and Scale is not clear enough. The two methods are introduced and only empirically compared, and it seems that Sample mostly dominates Scale (according to Table 2).  Looking at the overall algorithm, both techniques look similar in terms of efficiency. Then when should one use Scale?

W5) Also, a natural approach seems to be combining Sample and Scale where the sampling and clip norms are both adjusted. Why not use this combination?

W6) The notion of individualized privacy ($\epsilon_p$, $\delta$)-DP seems to be confused with the original ($\epsilon$, $\delta$)-DP. This confusion leads to other confusions, e.g., whether ($\epsilon_1$, $\delta$)-DP in Theorem 3.1. and ($\epsilon_P$, $\delta$)-DP in Theorem 3.2. mean individual privacy or conventional DP.

W7) The experimental setup of only using up to three privacy budgets seems limiting. What happens if there are many more than three budgets?

W8) Some reference number is wrong. For example, in the last paragraph of the proof of Theorem G.1, Mironov et al. is [21] or [22], not [20].

**Questions:**

Q1) What is the actual name of ({$\epsilon_1$,...,$\epsilon_n$}, $\delta$)-DP and ($\epsilon_p$, $\delta$)-DP? Are they Individualized Privacy or something else? If they are Individualized Privacy, why did the authors change the term “Personalized DP” to “Individualized Privacy” even though you referred to the Personalized DP paper?

Q2) How can getSampleRate function be equivalent to the function get_noise_multiplier in Opacus? getSampleRate seems to return sample rate with input argument noise multiplier, while get_noise_multiplier returns noise_multiplier sigma.

Q3) The comparison between DP-SGD and the proposed algorithms is as expected because DP-SGD assumes the same privacy level over all data points. Are there any results related to computational cost?

Q4) A detailed description of the properties of each Sample and Scale is lacking. Sample seems to be superior overall in performance. Is there any reason to introduce the Scale technique? Or, what are the advantages of the Scale method compared to the Sample method?

Q5) What is Thm. 4 and Thm. 11 from Mironov et al. [21] in the proof of Theorem G.1.? I could only find Proposition and Lemma in the paper. Did you mean [22]?

Q6) How can you simply prove Theorem G.1. and Theorem G.2. without proving some basic theorems or properties of newly designed privacy notion? I don't think previous DP techniques can be applied to individual privacy directly. The authors just directly use the conversion technique from RDP to ($\epsilon$,$\delta$)-DP of previous works as a technique for the individual privacy conversion even though the privacy notion is new.


**Limitations:**

The authors discuss limitations.

---

> ### Author Rebuttal · Authors · 2023-08-09
>
> > **W1 & Q6: Application of SGM theorems to IDP**
>
> The proof in G.1 showing that the entire mechanism satisfies $(\{\varepsilon_1, \varepsilon_2, \dots, \varepsilon_P\}, \delta)$-IDP is built on the observation that our methods can be considered as $P$ simultaneously executed SGMs that update the same model: The privacy groups divide our training data in $P$ disjoint subsets. Each of the $P$ SGMs operates on a different partition of the training data, using different individual sample rates or clip norms.
> Given that we perform the privacy accounting also on a per-group basis, each privacy group is just an original SGM (with different privacy parameters). Hence, the original SGM theorems apply.
>
> > **W1: Implementation of IDP-SGD**
>
> We build our implementation on top of Opacus.
>
> **Code flow**: We extend the PrivacyEngine to additionally take per-point budgets, i.e., an epsilon value for each training data point and to derive the adequate privacy parameter for each point (sample rates/clip norms). We also added an IndexedDataset to Opacus to be able to refer to individual data points via their index.
>
> **Individual sample rates**: We implement a data loader based on a custom weighted sampler. Our sampler adapts the functionality of Opacus’ UniformWithReplacementSampler and also uses the torch.rand function. In contrast to the UniformWithReplacementSampler that compares the generated random values against a uniform threshold to determine which data points will be in the current mini-batch, we compare point-wise, and each data point has an individual threshold that depends on its privacy assignment.
>
> **Individual clip norms**: The standard private optimizer in Opacus separately clips the gradient of each data point within a mini-batch. For our Scale, during clipping, each data point is clipped according to their privacy budget (with respective clip norm).
>
> The above description is reflected in the code attached to this submission.
>
> > **W1: Algorithm 3**
>
> We made space for Algorithm 3 in the main paper by merging Figure 3a) and 3b).
>
> > **W2: Optimality of individual privacy parameters**
>
> Our code implements Algorithms 1 and 2 to derive privacy parameters such that all privacy groups’ budgets will be exhausted at the end of training. Copying the per-group final privacy consumption from the log file of a training with CIFAR10, first privacy setup, Table 2 we see that we achieve `Sample: [0.997, 1.995, 2.992]`, and `Scale: [0.998, 1.995, 2.994]` in practice with eps=1,2,3 specified. I.e., the actual privacy consumption is tight. When adjusting the precision parameter in our calculation of privacy parameters from 0.0001 (displayed results) to e.g. 0.00001, values even closer to 1,2,3 can be achieved.
>
>
> > **W3&Q2: get_noise_multiplier vs. get_sample_rates**
>
> We meant that these functions are *conceptually* equivalent. They both rely on an interactive process where one privacy parameter is to be found in a binary-search style while the others are fixed.
>
> > **W4&Q4: Comparing Sample and Scale**
>
> We report the two best performing methods in the main paper (the other methods are in Appendix E).
>
> We observe two scenarios in which Scale is superior to Sample. (1) For full-batch gradient descent [9], Scale can be used while Sample is not applicable, given that all data points are sampled. (2) For extremely large dataset sizes (e.g. >500k), and very small mini-batch sizes (e.g. 32), due to rounding of the extremely small sample rates, the rates for two different privacy groups can become very similar or the same, preventing a fine-grained individualized privacy. This does not happen for Scale, where the clip norms are independent from the number of data points. The results in the general response show that, e.g., on BERT for SNLI Scale outperforms Sample.
>
> > **W5: Combining Sample and Scale**
>
> The combination of Sample and Scale extends the number of hyperparameters and potential to find better performing algorithms. It can be implemented as follows: obtain individual sampling probabilities as in SAMPLE. Once a mini-batch is sampled, use the respective data points’ indices and clip their gradients as in SCALE.
>
> > **W6: Notation for IDP**
>
> We would like to clarify that ($\epsilon_1$, $\delta$)-DP in Theorem 3.1. and ($\epsilon_P$, $\delta$)-DP in Theorem 3.2. mean conventional DP.   Semantically, $\epsilon_P$, $\delta$-DP expresses that  the epsilon value $\epsilon=\epsilon_p$ is equal to the privacy budget of the group with the highest privacy budget from individualized DP.
> To avoid confusion with the notation, we updated the paper to use the abbreviation $(\{\epsilon_1, \epsilon_2, \dots, \epsilon_P\}, \delta)$-**I**DP or ($\epsilon_P$, $\delta$)-**I**DP when we refer to individualized privacy, and ($\epsilon{(_P)}$, $\delta$)-DP for standard DP.
>
> > **W7: More privacy groups than 3**
>
> We already present such a result in Figures 5, 6, and 7. In these setups, we have *10* different privacy groups. To show the flexibility of our methods, we ran additional experiments on the MNIST dataset, using 100 evenly sized privacy groups with budgets [1, 1.05, 1.1, ..., 5.9, 5.95] and report the results in the general response.
>
> > **Q1: Naming “Individualized”-DP**
>
> There are different names for this concept. We also cite [2] who refer to the concept as heterogeneous differential privacy. We use Individualized DP-SGD following the naming of Individualized PATE [5].
>
> > **Q3: Computational costs**
>
> We performed experiments measuring runtimes. The detailed results can be found in the general answer. We observe that Sample and Scale do not significantly increase runtime in comparison to standard DP-SGD with (406.33 sec for Sample, 414.33 for Scale vs. 403.67 for DP-SGD).
>
> > **W8&Q5: References**
>
> Yes, we meant [22]. The full paper in the supplementary material accidentally is missing citation [7] from the submission. We fixed all the citations for the updated version of our paper.

---

> > ### Author Response · Authors · 2023-08-14
> > **Sharing Additional Results Regarding the Combination of Sample and Scale**
> >
> > To further illustrate how we addressed the point about combining our Sample and Scale methods, we implemented the combination that we described in the rebuttal.
> >
> > We allow to weight the different methods to arbitrary fractions (e.g., 50% Sample, 50% Scale), and derive the privacy parameters such that all privacy groups exhaust their respective budgets at the end of the specified number of iterations.
> >
> > We ran additional experiments to showcase the advantage of combining both methods. We used the MNIST dataset and the first privacy setup (0.34, 0.43, 0.23) with privacy budgets {1,2,3} and weighted our methods as indicated in the table below.
> >
> >
> > | Sample (weight in %) | Scale (weight in %) | Test Accuracy (in %) |
> > |------------|-----------|---------------|
> > | 0         | 100      | 97.78+-0.08   |
> > | 25        | 75       | 97.75+-0.10   |
> > | 50        | 50       | 97.80+-0.10   |
> > | 75        | 25       | 97.80+-0.10   |
> > | 100       | 0        | 97.81+-0.09   |
> >
> > The baseline (first and last rows) are taken from Table 2 in the original submission, while the additional three rows in between with the new results represent average test accuracy and the standard deviation over ten independent random runs over different weights (w) assigned to the respective method..
> >
> > Our results demonstrate a clear progression from the lower performing Scale approach to the better performing Sample. This aligns with expectations based on our implementation, where we combine the methods based on the parameters found for them sequentially.
> >
> > Note that each of our methods has its advantages and disadvantages and the combination of both methods provides more options to balance (dis)advantages. For example, when some points’ sample rates are extremely small, then the model might never see those points during training and the combined method could reduce this possibility while still improving over the Scale-only method.

---

> > > ### Comment · Reviewer_Sruy · 2023-08-15
> > >
> > > Thanks for thoroughly addressing my concerns including extensive experiments. I am increasing my score to be on the positive side.

---

> > > > ### Author Response · Authors · 2023-08-15
> > > > **Thank you to the reviewer**
> > > >
> > > > We would like to thank the reviewer for engaging in the discussion with us and for increasing their score.

---

### Official Review · Reviewer_7px6 · 2023-07-08

**Soundness:** 3 good
**Presentation:** 4 excellent
**Contribution:** 3 good
**Rating:** 6
**Confidence:** 3

**Summary:**

This paper proposed two variants of DP-SGD by manipulating the sampling rate and the gradient clipping bound for different groups to achieve the goal of having different privacy budgets for those groups and improving the overall performance of DP-SGD. The authors proved theoretical privacy guarantees for both variants. They also conducted an extensive experimental evaluation to showcase the advantage of their methods in many aspects.

**Strengths:**

1. The authors considered an important problem, having different privacy budgets for different groups of people to boost the overall utility, and proposed two algorithms to achieve the goal. The privacy guarantee of the algorithms is proved. Both algorithms are easy to implement and look novel although intuitive to me.
2. The experiments in this paper are adequate for the evaluation of the proposed methods. The results are well-organized and visualized, which supports the validity and the advantage of the methods, and the details are provided enough for reproducibility. I understand and appreciate that the authors put most of the valuable experimental results in the appendix due to the page limitation.
3. The writing and the organization of the content are good.

**Weaknesses:**

1. The composition theorem for Algorithms 1 and 2 is not proved or discussed in this paper. The proofs of Theorem G.1 and G.2 look okay to me although I think more explanations would make it more friendly to the general audience. However, I think there is still a need to show the composition theorem similar to the results in Feldman and Zrnic [9]. Moreover, if it is not much work for rewriting related parts of the paper, I think a more strict way to account for the privacy budget usage is by the Rényi Filter [9], i.e., individual RDP instead of RDP.
2. Section 4.2 uses membership inference to empirically verify the privacy protection for the claimed DP guarantee, but the results in Figure 2 do not explicitly show the protection separately for each group ($\varepsilon=10$ and $\varepsilon=20$) compared with the DP guarantee. The authors may show the curves under two additional settings: vanilla DP-SGD with $\varepsilon=10$, and vanilla DP-SGD with $\varepsilon=20$. Then compare the two solid curves in Figure 2 with the two additional curves respectively.
3. Both Algorithms 1 and 2 will potentially change the objective function of SGD (although DP-SGD with the gradient clipping has already changed it). There is no discussion on how specific choices of privacy budget distribution will affect the convergence and final performance of the optimization. This is important since it is related to the privacy-utility tradeoff. Please see my comment in *Limitations*.


Minor weaknesses:
1. Line 66. I think Table 7 is the same as Table 2 which is in the main text, not the appendix.
2. Line 86. $\log 1/\delta$ -> $\log(1/\delta)$
3. Line 98, equation (2). The inequality holds with many constraints on $\alpha$. See Theorem 11 in Mironov et al. [21]
4. Algorithm 1. The authors wrote: it is equivalent to Opacus' function get_noise_multiplier while Algorithm 2 is also equivalent to the same function. I think there is a typo.
5. Algorithm 1, Line 5. I think this is more like a grid search which always decreases $\sigma_{\mathrm{sample}}$, while a binary search would be better.
6. Line 209. The notation $\stackrel{\text{!}}{=}$ is not defined.
7. Line 767 in Section G.2. $\sum_{L\subset D}$ -> $\sum_{L\subset D'}$
---
I have read the rebuttal which addressed my questions and the weakness concerns.

**Questions:**

1. In Line 379, could you explain the meaning of 'batch gradient descent'? Is it different from SGD? If yes, maybe it can be replaced with the word 'full-batch gradient descent' or just 'gradient descent'.
2. In Table 8, we can see that for the first two rows, the accuracy decreases for classes 1-9 with a higher privacy budget of class 0 compared with the results with the same privacy budget of class 0. The same phenomenon exists in other rows. Intuitively, the higher budget of other classes should provide more information for the classifier so that the accuracy for each group would increase. Could you provide more insight into why this intuition is incorrect under this setting?

**Limitations:**

1. The discussion of the incentive and consequence of people choosing weak privacy protection is not enough in the main text. I found the content in Section D.2 very interesting since it aligned with my intuition: the more privacy being sacrificed, the better utility could potentially be obtained. Also, the results in Table 8 show that having a group with a higher privacy budget will affect the utility of other groups, which could lead to problems like the Prisoner's dilemma where people in the dataset have to give a higher privacy budget in order to maintain their original utility given the fact that other people are willing to have a higher privacy budget.

---

> ### Author Rebuttal · Authors · 2023-08-09
>
> We thank the reviewer for their comments and suggestions and provide our answer below:
>
> >**W1: Composition theorems**
>
> We would like to point out that our algorithms have the exact same composition (within every privacy group individually) as the standard DP-SGD and do not require additional theorems on composition. In detail, our contribution is to propose a novel way of individual privacy **assignment**. Our privacy accounting follows the standard accounting also used within DP-SGD with the difference that accounting is performed within a privacy group separately. Feldman and Zrnic [9]’s contribution, in contrast, is to propose a novel individual privacy **accounting** mechanism which then requires a different composition. Their accounting differs from standard DPSGD accounting in the sense that it does not account for worst-case privacy leakage, but that it estimates the ‘actual’ privacy leakage. The composition from standard DP-SGD is, thereby, not directly applicable for them while it is for us.
>
> Finally, we would like to note that in Section 5, we show that our individualized privacy assignment can be integrated into the individualized privacy accounting by [9]. We do so by assigning individual privacy budgets to different data points and then rely on the individual privacy accounting by [9] (using their composition theorem).
>
> >**W2: MIA for two additional settings**
>
> We thank the reviewer for this suggestion. We performed this experiment, trained a model with $\varepsilon=10$ and another one with $\varepsilon=20$ using standard DP-SGD on CIFAR10. Then, we performed a LiRA MIA attack with 512 shadow models (the same as in Figure 8a and 8b in the submission) and report the detailed results in the general response. The resulting AUCs can be summarized as follows:
> ||All points ($\varepsilon=10$ and $\varepsilon=20$)|Points with $\varepsilon=10$|Points with $\varepsilon=20$|
> |-----------------------------|------------------------------------------------------|-------------------------------|------------------------------|
> |Sample|0.56|0.537|0.581|
> |Scale|0.571|0.552|0.589|
> |DP-SGD ($\varepsilon=10$)|-|**0.568**|-|
> |DP-SGD ($\varepsilon=20$)|-|-|**0.59**|
>
> Our key observations are:
> 1. IDP-SGD reduces the privacy risk for the data points with $\varepsilon=10$ (in standard DP-SGD, their AUC=0.568 which is higher than with Sample 0.537 or Scale 0.552).
> 2. This privacy gain does not come at large expense of points with $\varepsilon=20$ whose privacy risk remains roughly the same (AUC=0.59 in DP-SGD vs Sample 0.581 and Scale 0.589).
>
> > **W3: Objective function, convergence, and final performance**
>
> We would like to point out that already DP-SGD itself (especially with the gradient clipping) does not offer rigorous convergence guarantees except in convex, Lipschitz, or Lipschitz-like regimes. For this reason, it is a common practice in work on DP-SGD to rely on empirical evaluation for the utility, see for example [Differentially private learning needs better features (or much more data)](https://arxiv.org/abs/2011.11660), [Unlocking high-accuracy differentially private image classification through scale](https://arxiv.org/pdf/2204.13650v2.pdf).
>
> Following this common practice, we experimentally evaluate a wide range of privacy distributions and different privacy parameters (e.g., two distributions with eps=1,2,3 in Table 2, a distribution with eps=10,20 in Section 4.2, another distribution with eps=1,2,3 in Table 8 and Figure 5+6, and a distribution with ten different epsilon in range between 0.5 and 6.1 in Table 9).
> In the general response, we also provide a new experiment with 100 different privacy groups.
> Over all experiments, we increase the utility of DP-SGD and do not observe any convergence issues in our experimental results.
>
> > **Minor comments**
>
> We thank the reviewer for carefully reading our paper. We removed Table 7 and line 66 now refers to Table 2. We also corrected the other points mentioned by the reviewer and explain the notation !=, which refers to “should be equal to” in the updated version of the paper.
>
> > **Q1: Batch gradient descent**
>
> We indeed refer to full-batch gradient descent and adopted the reviewer’s suggestion on using this term within our work.
>
> > **Q2: Utility increase and decrease due to individualized privacy**
>
> We thank the reviewer for their careful study of Table 8 and are happy to provide further insights into the observed phenomenon:
> What we observed in our experiments is that the group that uses a higher privacy budget (e.g. group class 0) provides more information to the model and makes the model learn the features of this particular group better (reducing the model’s performance on other groups’ features). We think that this phenomenon is illustrated very interestingly, for example, in Figure 6a). We observe that if class 9 gets the higher privacy budget (last column), the class that suffers the largest utility loss is class 4 (which is the most similar digit to the 9 and suffers from the model paying more attention to the features of the 9). This shows that when the privacy groups differ significantly in their distributions, by increasing the privacy budget of one group, especially this group will benefit from utility improvements.

---

> > ### Comment · Reviewer_7px6 · 2023-08-16
> > **Thank you for the response!**
> >
> > W1. Now I understand the composition results in this paper.
> >
> > W2. This table looks good to me.
> >
> > W3. Yes. I understand the difficulty here. My concern is more about the characterization of the utility loss for groups with lower privacy budgets. If the convergence (e.g., the decrease of train loss) is slower for the groups with lower privacy budget, then DP-SGD may need a longer time to converge which also affects the choice of the number of iterations of SGD. I understand that this may be out of the scope of this paper but still raises my concern.
> >
> > Based on your explanation for W1, I have increased my score from 5 to 6.

---

> > > ### Author Response · Authors · 2023-08-16
> > > **Thank you to the reviewer**
> > >
> > > We would like to thank the reviewer for their careful read of our response, for engaging in the discussion with us and for increasing their score.

---

> > > ### Author Response · Authors · 2023-08-19
> > > **Sharing additional Results regarding the Training Dynamics and Loss History of the privacy group with smallest epsilon**
> > >
> > > To address the reviewer’s concern regarding the loss history of the group with the smallest privacy budget ($\varepsilon=1$), we ran additional experiments.
> > >
> > > We collected the loss values (and accuracy values) over the 4 following setups on the MNIST dataset and privacy setup 2 (54%,37%,9%, $\varepsilon$={1,2,3}):
> > > - loss value for privacy group with $\varepsilon=1$ for the Sample method;
> > > - loss value for privacy group with $\varepsilon=1$ for the Scale method;
> > > - training with standard DP-SGD ($\varepsilon=1$) solely on the 54% of data points that have the privacy budget of $\varepsilon=1$, and
> > > - training with the standard DP-SGD ($\varepsilon=1$) on 100% of the MNIST dataset.
> > >
> > > In the following table, we depict the training loss of the privacy group with $\varepsilon=1$ during training:
> > > | **Epochs** | **Sample, $\varepsilon$=1,2,3 (loss of group eps=1)** | **Scale, $\varepsilon$=1,2,3 (loss of group eps=1)** | **DP-SGD 54%, $\varepsilon$=1** | **DP-SGD 100%, $\varepsilon$=1** |
> > > |------------|---------------------------------------------|--------------------------------------------|-----------------------|------------------------|
> > > | 20         | 0.145                                       | 0.148                                      | 0.211                 | 0.162                  |
> > > | 40         | 0.116                                       | 0.121                                      | 0.187                 | 0.143                  |
> > > | 60         | 0.109                                       | 0.115                                      | 0.183                 | 0.142                  |
> > > | 80         | 0.105                                       | 0.112                                      | 0.184                 | 0.142                  |
> > >
> > > After training finishes, this leads to the following accuracies for the respective group:
> > > |                   | **Sample, $\varepsilon$=1,2,3** | **Scale, $\varepsilon$=1,2,3** | **DP-SGD 54%, $\varepsilon$=1** | **DP-SGD 100%, $\varepsilon$=1** |
> > > |-------------------|-----------------------|----------------------|-----------------------|------------------------|
> > > | Test Accuracy (%) | 97.43                 | 97.26                | 95.36                 | 96.56                  |
> > >
> > >
> > > Our results indicate that the availability of more information from the data of different privacy groups in our Sample and Scale methods enables the model to learn better features. This leads to lower loss for the data points with privacy budget $\varepsilon=1$ than if the model was trained on only those 54% of points. The effect of including more data (i.e., using 100% of the MNIST dataset), all with $\varepsilon=1$, is weaker, indicating that even the data with highest privacy requirement benefits substantially from our individualized privacy in terms of reduced loss and increased accuracy.

---

### Official Review · Reviewer_3QXJ · 2023-07-09

**Soundness:** 3 good
**Presentation:** 3 good
**Contribution:** 3 good
**Rating:** 6
**Confidence:** 5

**Summary:**

This paper proposes two variants of Differentially Private Stochastic Gradient Descent (DP-SGD) to train machine learning models that satisfy approximate personalized differential privacy (PDP), following the definition of Jorgensen et al. [15]. In contrast to vanilla DP-SGD where all points in the training dataset are assigned a global, uniform privacy budget $\varepsilon$, the authors consider a scenario where points are split into privacy groups $\{\mathcal{G}_1,\ldots,\mathcal{G}_P\}$ and each group $\mathcal{G}_p$ is assigned a different privacy budget $\varepsilon_p$. This allows individuals who contribute training data to specify their privacy preferences, reflecting the variety of privacy attitudes among users observed in past surveys.

To achieve PDP the paper modifies DP-SGD in one of two ways:

1. **Sample** method: adjusting the sampling rate used to construct mini-batches, so that points in groups with a higher privacy budget are sampled more often than points in groups with lower privacy budget (and thus, stringent privacy preference).

2. **Scale** method: adjusting the noise multiplier to scale the variance of noise added in a per-point basis. Rather than doing this directly, which would be inefficient, the method achieves the same effect indirectly by scaling the clipping norm of per-point gradients.

These modifications are such that given target group privacy budgets and a number of training steps, the mini-batch noise multiplier in DP-SGD can be chosen so that the privacy budget for each group is exhausted at the end of training, maximizing utility.

The paper evaluates **Scale** and **Sample** on CNNs trained over MNIST, SVHN and CIFAR-10 using 3 privacy groups with budgets $\varepsilon_p = p$ for $p \in \lbrace1,2,3\rbrace$, distributing points into groups according to proportions used in prior work. This evaluation shows top-1 accuracy improvements ranging between 1-5% over vanilla DP-SGD with a uniform privacy budget $\varepsilon_1 = 1$ (which guarantees PDP, but underutilizes the budget for the other two groups). The methods also compare favorably to Individualized PATE [5], which uses different techniques but achieves the same privacy guarantee.

The paper also demonstrates that the empirical evaluation of membership inference attacks against models trained with PDP using the above two methods shows the expected gap between groups with different privacy budgets. Finally, the authors discuss how to combine individual privacy assignment with individual privacy accounting as done by Feldman and Zrnic [10] for batch gradient descent.

The supplemental material provides implementations of **Scale** and **Sample** in the Opacus library as well as scripts to reproduce the results in the paper.

**Strengths:**

1. **First DP-SGD variants designed to achieve personalized differential privacy**
There is abundant literature for adapting differentially private mechanisms for general data analysis to achieve personalized differential privacy as well as for computing personalized differential privacy guarantees for training data points in ML models. While PDP accounting alone can be used to improve model utility modestly (see Feldman and Zrnic [10]), as far as I know the only other method that trains models with individual privacy assignments is Individualized PATE [5] which uses significantly different techniques.

2. **Original combination of existing mechanisms**
The **Sample** mechanism is inspired by the "Sampling" mechanism of Jorgensen et al. [16], while **Scale** is inspired by the "stretching" mechanism of Alaggan et al. [2]. Combining these mechanisms with DP-SGD is relatively straightforward, but has not been done before.

3. **Problem is well motivated and solution is contextualized relative to prior work**
The authors discuss related work up front, explain how their solution fits into the existing literature, and justify their contributions.

4. **Complete algorithmic descriptions of proposed methods** The paper describes the adaptations of DP-SGD in sufficient detail, including pseudocode and justifications for design decisions.

5. **Demonstrates utility gains compared to alternative solutions** The empirical evaluation on image classification tasks demonstrates a modest utility gain compared to alternative baselines.

6. **Comprehensive supplemental material** The Appendix includes valuable additional discussions. The accompanying code is a significant extension to an existing library and demonstrates the practicality of the techniques.


**Weaknesses:**

1. **Limited empirical evaluation** The empirical evaluation is limited to convolutional architectures trained from scratch for image classification. It is unclear how the paper findings hold for other architectures, modalities, tasks, or practically relevant scenarios, such as fine-tuning.

2. **Modest utility gains** The utility gains with respect to vanilla DP-SGD are modest (1-5%). The comparison to alternative baselines with simpler implementations in Appendix E shows even more modest gains. For instance, simply training sequentially on each group separately using DP-SGD achieves accuracies only 0.3-0.41% below the best of **Sample** and **Scale** (see Table 12). It is unclear to what extent these differences are statistically significant, whether comparable effort has been put in hyperparameter tuning, and whether mild modifications to this alternative baseline could outperform **Sample** and **Scale**.

3. **Unsupported claims about the confidentiality of privacy preferences** The authors argue that privacy preferences (i.e., per-group privacy budgets) should be kept private and claim that their modifications of DP-SGD do not leak information about privacy preferences because untrusted parties only interact with the final trained model. This claim is unsupported by proof (compare this to Allagan et al. [2], which prove a similar guarantee explicitly) and is made with respect to a different adversary model than the privacy guarantees of training data in DP-SGD, which consider adversaries that observe noisy gradients released at every training iteration, not only the final model.


## Minor comments

- Given the number of references to numbered lines in Algorithm 3, it would be convenient to include the listing in the body of the paper.

- l.76: "[D]iffer in any one record" is ambiguous. Consider making clear that you use the add/remove neighboring relation. That is, $D,D'$ are adjacent if $D = D' \cup \{x\}$ for some record $x$, or vice versa.

- l.79: "DP bounds privacy leakage for any individual". This is only true under the assumption that an individual contributes at most one record.

- l.86: The original conversion from $(\alpha, \rho)$-RDP to $(\varepsilon, \delta)$-DP of Mironov [21] is suboptimal. Balle et al. [A, Theorem 21] provide a tighter conversion (which is the one Opacus uses).

- Table 1: $\sigma_{\rm SAMPLE}$ should read $\sigma_{\rm sample}$.

- Algorithm 1: Missing inputs: scaling factor $s_i$, target sample rate $q$.

- Algorithm 1: In l.1, you could initialize $\sigma$ using $\textit{getNoise}$ as $\sigma \gets \textit{getNoise}(\varepsilon_1, \delta, q, I)$.

- Algorithm 1: In l.4 $G_p$ should read $\mathcal{G}_p$.

- l.190: "our Sample [method]"

- l.209: While I have seen it being used before, I think that the notation $\stackrel{!}{=}$ is far from being conventional.

- l.215: "[W]e still require the expected average sampling rate to remain $q$ to obtain constant mini-batch size $B$". As you said just before, the mini-batch size isn't constant, $B$ is the **expected** size. Rather, you want the expected mini-batch size in IDP-SGD to be the same as in vanilla DP-SGD with sampling rate $q$.

- l.225: There's an extra closing parenthesis.

- l.232: DPSGD should read DP-SGD.

- l.260: Use \{$\sigma_1,\ldots,\sigma_P$\} rather than $\sigma_p$-s.

- l.313 (and subsequently): "Lira" should read "LiRA"

- Numbers in bibliographic references are shifted between the paper and the full paper (with Appendix) provided as supplemental material.

- Appendix, Algorithm 3: In line 11, $\theta_T$ should be $\theta_I$.

- l.566: $\sigma_{\rm in}$, $\sigma_{\rm out}$ should read $\sigma^2_{\rm in}$, $\sigma^2_{\rm out}$, respectively.


- l.576: "Line 3 in Algorithm 3" should be "Line 8 in Algorithm 3"

- l.591-596: $G_p$ should read $\mathcal{G}_p$.

- l.691: "Appendix E.3" should read "Table 12"

- Table 10: How can it be that you get a statistically significant negative $\Delta$ for one of the models? That would reverse the relation between the two privacy groups.

**Questions:**

1. **Confidentiality of privacy budgets**
The paper argues that IDP-SGD preserves the confidentiality of group privacy preferences, at least against parties that only interact with the final trained model. However, intuitively, models trained with a uniform privacy budget of $\varepsilon=1$ and models trained on the same data but
increasing the privacy budget for a single group from $\varepsilon=0.1$ to $\varepsilon=100$ will likely perform noticeably different on that group. Can you provide a proof that IDP-SGD satisfies some form of confidentiality for privacy preferences that makes explicit the adversary model and assumptions?

2. **LiRA AUC scores**
Carlini et al. [6] report LiRA results on models trained with DP-SGD and $\epsilon = 8$ (not far from one your choices of $\epsilon = 10$), with a significantly lower AUC score of 0.503 compared to the score of 0.537 that you report. Even for $\varepsilon > 5000$, their best result has a lower AUC score of 0.527. Could you explain this difference?

**Limitations:**

The authors adequately address societal impact but only some limitations.

A paragraph at the end of the introduction discusses the need for openly communicating privacy risks and regulatory supervision. This points to Appendix A, which discusses broader impacts in the light of prior studies and some limitations: evaluation on standard benchmarks with simulated privacy preference distributions; practical impact measured via membership inference attacks only. I think that the limited nature of the evaluation on CNN models trained from scratch on visual tasks should also be brought up front, as the results might differ in other settings.

---

> ### Author Rebuttal · Authors · 2023-08-09
>
> First of all, we would like to express our gratitude to the reviewer for their thorough review and detailed feedback that goes beyond any expectation.
>
> > **Limited empirical evaluation**
>
> We thank the reviewer for their suggestion to extend our experimental evaluation on other architectures, tasks, modalities, and fine-tuning. We ran additional experiments with IDP-SGD for:
> - fine-tuning a BERT transformer on the SNLI dataset for natural language inference,
> - training a ResNet 18 with IDP-SGD from scratch on the CIFAR10 dataset for image classification,
> - training a linear embedding model on the IMDB dataset for text classification,
> - and training a character-level RNN from scratch to classify surnames to languages.
>
> The results are included in the general response.
> Over all the considered experiments, IDP-SGD outperforms standard DP-SGD which underlines that our findings on the performance improvements of IDP-SGD generalize over different setups.
>
> > **Modest utility gains**
>
> The main goal of our work is to enable individually private training with different epsilon values. We compare the possible options and present the best performing method in the main body of the work. We performed thorough hyperparameter tuning over all our methods, including E.1 and E.2. Note that the hyperparameter tuning for E.1 and E.2 goes beyond standard parameters. For E.1, one also needs to tune over which group(s) to exclude, and for E.2 in what order to train on the different groups. We performed all these analyses and reported the best results.
>
> From a conceptual viewpoint, Scale and Sample outperform the other baselines when the data distribution between the different privacy groups differs significantly. E.1 which completely excludes whole groups from training will not be able to learn the high-privacy groups at all, while E.2 which trains privacy groups sequentially will suffer from catastrophic forgetting where the final model will perform worse on the groups first seen during training. In contrast, in Sample and Scale, all privacy groups contribute to the training over the entire training process.
>
> Finally, we would like to note that in DP-SGD research, a utility improvement of 1-2% is a significant contribution, especially in the low-privacy regime (eps=1,2,3) that we are operating in. See for example, the [[Unlocking high-accuracy differentially private image classification through scale]](https://arxiv.org/pdf/2204.13650v2.pdf)-paper that outperforms prior work by less than 2% at epsilon=3. Looking into Fig 1a) of the paper, we see that the improvement of the other prior baselines over each other is usually in the range of 1-2%.
>
> > **Minor comments**
>
> We made space for Algorithm 3 in the main paper by merging Figure 3a) and 3b), clarified the DP definition, and fixed all the formatting suggestions pointed out by the reviewer. Additionally, we replaced the conversion from RDP to (eps,delta)-DP by the one proposed in Balle et al. Finally, we adapted Algorithm 1.
>
> Regarding the negative delta in Table 10, we are sorry for the confusion. The delta (effect size) in the table should be positive 4.03. We corrected the typo.
>
> > **Question 1: Confidentiality of privacy budgets**
>
> We wrote a thorough explanation on the confidentiality of privacy budgets in our methods in the general response.
>
> Regarding confidentiality of privacy values in [2], we would like to mention that their application of HDP is within a distributed gossip-based semantic clustering protocol that operates in an iterative manner. In the protocol, pairs of users calculate the cosine similarity between their preferences (see page 11). The Stretching mechanism used for this relies on both users’ privacy preference vectors as parameters. This means, the users get to directly interact with each other’s privacy preference vectors–requiring privacy protection for these vectors. In contrast, in our setup, all privacy-preference based operations are already performed when the end user (potential adversary) gets to interact with the model where determination of privacy budgets is difficult and highly impractical (see argument 1. and 2. in the general response).
>
> When dealing with different privacy groups, [2] implements a form of deniability for users’ membership in a certain privacy group by uniformly sampling the privacy values over groups in an overlapping way: the unconcerned sample from {1.}, the pragmatists from {0.5, 0.75, 1.}, and the fundamentalists from {0, 0.5, 1}, such that on average, the values will be 1., 0.75, and 0.5 within the groups (0 denotes the strongest, while 1 is the weakest privacy guarantee). Such an approach can be implemented into our IDP-SGD framework as well. The drawback is that an individual who expresses strong privacy preferences (fundamentalists) might actually be assigned a privacy value 1 by randomness, which causes the same privacy leakage as for an unconcerned individual. Depending on the individuals’ preferences and the application, when protecting privacy group membership is more important than protecting the actual data, this approach might still be valid.
>
> We added a discussion about adopting this approach into our paper as Section 3.5.
>
> > **Question 2: LiRA Scores**
>
> We thank the reviewer for this insightful detailed question. The differences in our vs. their setup that can cause our higher AUC score are:
> 1. We set eps=10 for half of the data and eps=20 to the other half, which is significantly higher than their eps=8.
> 2. The accuracy of our models differs from theirs. Our target model achieves a train accuracy of 68.46% on its 25,000 member data points, and a test accuracy of 64.89% on its 25,000 non-member data points. They report 61.3% test accuracy (we could not find their train accuracy in the appendix). Differences in accuracy, especially differences in the train-test gap can cause differences in the MIA success rate.
> 3. Their attack on CIFAR10 uses 256 shadow models while we use 512.

---

### Author Rebuttal · Authors · 2023-08-09

We would like to thank all reviewers for their feedback which has greatly helped us improve the paper. We are glad that the reviewers recognize our work to address an important problem by proposing the first personalization mechanisms for DP-SGD (3QXJ) which are novel, easy to implement, and provide theoretical privacy guarantees (7px6, Sruy, r49G). Our methods ‘yield a modest utility over alternative baselines’ (3QXJ), (Sruy,r49G), and the ‘experiments in this paper are adequate for the evaluation’ with additional valuable results and discussions in the appendix (7px6). The reviewers appreciate the writing and the organization of the paper (7px6, r49G) and we hope that this work can contribute to making machine learning with individualized privacy guarantees practical. Below we offer clarifications to some common questions.

>**1. Extending experiments**

**Other (larger) architectures, modalities, tasks, and fine-tuning:**
Following reviewers 3QXJ and r49G, we evaluated IDP-SGD in a broader scope and
fine-tuned a BERT transformer (bert-base-uncased) on the SNLI dataset for natural language inference,
trained a ResNet 18 from scratch on the CIFAR10 dataset for image classification,
trained a linear embedding model *(one embedding and two linear layers)* on IMDB data for text classification,
and trained a character-level RNN *(DPLSTM, three layers, hidden size: 128)* from scratch to classify surnames to languages.

|Model|Dataset|DP-SGD (eps=5)|Sample (eps=5,7.5,10)|Scale (eps=5,7.5,10)|
|-----------|----------|----------------|--------------------|-------------------|
|BERT|SNLI|75.91+-0.23|76.11+-0.21|76.5+-0.17|
|ResNet18|CIFAR10|47.52+-0.84|48.52+-0.69|48.77+-0.73|
|Embedding|IMDB|72.69+-0.27|73.27+-0.3|73.34+-0.11|
|||**(eps=1)**|**(eps=1,2,3)**| **(eps=1,2,3)**|
|RNN|Surnames|60.86+-0.78|65.56+-0.96|66.0+-1.19|
Over all setups, our IDP-SGD outperforms standard DP-SGD → see also attached PDF, Table 1.

**Runtimes:**
Inspired by reviewer Sruy, we analyzed runtimes of our methods. E.g., for the first privacy setting of Table 2, and the CIFAR10 dataset with the CNN architecture, we observe that Sample and Scale do not significantly increase runtime over DP-SGD with 406.33 sec for Sample, 414.33 for Scale vs. 403.67 for DP-SGD → see PDF Table 3.

Regarding the time for computing the privacy parameters which takes place *exactly once* before the training with a new privacy setup, we included a detailed Table 4 in the PDF.  Computation time depends on a precision parameter and the number of groups: For example, with 8 groups and precision=0.0001 (used in this work), Sample requires 20 sec. and Scale 5 sec. to determine the parameters.

**More privacy groups:**
Following reviewer Sruy, we implemented IDP-SGD training on MNIST with 100 evenly sized privacy groups with budgets [1, 1.05, 1.1, ..., 5.9, 5.95]. Our methods achieved the following accuracy: Sample: 98.17%, Scale: 98.21% vs. the standard DP-SGD baseline with epsilon=1: 96.75%. → see PDF Table 2.

**Compare MIA with standard DP-SGD**

Given the suggestion by reviewer 7px6, we trained models with $\varepsilon=10$ and $\varepsilon=20$ with standard DP-SGD and compared the results to our MIA results with IDP-SGD. Our key observations are:
1. IDP-SGD reduces the privacy risk for the data points with $\varepsilon=10$ (standard DP-SGD: AUC=0.568; Sample: 0.537; Scale: 0.552).
2. This privacy gain does not come at large expense of points with $\varepsilon=20$ whose privacy risk remains roughly the same (AUC=0.59 vs Sample: 0.581 and Scale: 0.589).
 → see PDF Figure 1.

> **2. Confidentiality of privacy preferences**

The confidentiality of our privacy budgets results from the following observations:
1. Even for standard DP training, it was shown that one cannot perform sample-efficient black-box audits to determine the privacy-budget of a trained model [[Property Testing for Differential Privacy]](https://doi.org/10.48550/arXiv.1806.06427). Existing frameworks that perform auditing, e.g. [[Debugging Differential Privacy: A Case Study for Privacy Auditing]](https://arxiv.org/pdf/2202.12219.pdf) rely on training 1000-100,000 shadow models to get an estimate of the full model’s privacy guarantee.
In our IDP-SGD setup, the model does not have one single $\varepsilon$, but one $\varepsilon$ per group, aggravating the impracticability of auditing guarantees and limiting the applicability of existing frameworks. As a result, the adversary cannot determine the different per-group privacy budgets of the model.
2. Our MIA experiments in Section 4.2 show a difference in privacy risks over *entire privacy groups*. First note that these experiments (Figure 2) were the most costly ones in our paper.  We had to train more than 1500 (3 times 512) shadow models to generate the results. Yet, there are two limitations: a) The results only show differences in the groups, but do not reveal the respective epsilon values, b) they do not allow to perform per-point distinctions. This is because the distributions of privacy risks over the two groups still have a large overlap. Hence, given a single point’s risks and the two distributions, one cannot predict without error to what group the point belongs. We acknowledge that the error could be reduced when the privacy budgets between the groups differ significantly (0.1 vs. 100). Note however, that we suggest deploying the model in a setup where individuals simply specify their privacy preferences and the model builder or an ethics committee assign concrete epsilon values to these preferences. We observed that extreme differences between privacy groups’ $\varepsilon$s are less beneficial because they degrade overall model performance. Hence, in practice, these large differences are very unlikely. As a result, it will be very challenging for an adversary to determine which privacy group a data point belongs to. Even if they manage to, they will not be able to learn the privacy budget of the group (see 1.).

---

### Decision · Program_Chairs · 2023-09-21

**Decision:**

Accept (poster)

**Comment:**

Reviewers all generally agreed that this was an interesting and worthwhile direction. While the algorithms are not too surprising, it's a good first step. The weaknesses were not considered significant enough to warrant holding back this work for publication.